# Does the Rational Function Model's Accuracy for GF1 and GF6 WFV Images Satisfy Practical Requirements?

**Xiaojun Shan [1],\* and Jingyi Zhang [2]**

[1] Aerospace Information Research Institute, Chinese Academy of Sciences, Beijing 100094, China
[2] School of Science, China University of Geosciences (Beijing), Beijing 100083, China; 2119210055@email.cugb.edu.cn
\* Correspondence: shanxj@aircas.ac.cn

**Abstract:** The Gaofen-1 (GF-1) and Gaofen-6 (GF-6) satellites have acquired many GF-1 and GF-6 wide-field-view (WFV) images. These images have been made available for free use globally. The GF-1 WFV (GF-1) and GF-6 WFV (GF-6) images have rational polynomial coefficients (RPCs). In practical applications, RPC corrections of GF-1 and GF-6 images need to be completed using the rational function model (RFM). However, can the accuracy of the rational function model satisfy practical application requirements? To address this issue, a geometric accuracy method was proposed in this paper to evaluate the accuracy of the RFM of GF-1 and GF-6 images. First, RPC corrections were completed using the RFM and refined RFM, respectively. The RFM was constructed using the RPCs and Shuttle Radar Topography Mission (SRTM) 90 m DEM. The RFM was refined via affine transformation based on control points (CPs), which resulted in a refined RFM. Then, an automatic matching method was proposed to complete the automatic matching of GF-1/GF-6 images and reference images, which enabled us to obtain many uniformly distributed CPs. Finally, these CPs were used to evaluate the geometric accuracy of the RFM and refined RFM. The 14th-layer Google images of the corresponding area were used as reference images. In the experiments, the advantages and disadvantages of BRIEF, SIFT, and the proposed method were first compared. Then, the values of the root mean square error (RSME) of 10,561 Chinese, French, and Brazilian GF-1 and GF-6 images were calculated and statistically analyzed, and the local geometric distortions of the GF-1 and GF-6 images were evaluated; these were used to evaluate the accuracy of the RFM. Last, the accuracy of the refined RFM was evaluated using the eight GF-1 and GF-6 images. The experimental results indicate that the accuracy of the RFM for most GF-1 and GF-6 images cannot meet the actual use requirement of being better than 1.0 pixel, the accuracy of the refined RFM for GF-1 images cannot meet practical requirement of being better than 1.0 pixel, and the accuracy of the refined RFM for most GF-6 images meets the practical requirement of being better than 1.0 pixel. However, the RMSE values that meet the requirements are between 0.9 and 1.0, and the geometric accuracy can be further improved.

**Keywords:** GF-1 WFV image; GF-6 WFV image; geometric accuracy evaluation; rational function model; automatic matching

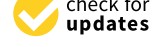

## 1. Introduction

The Gaofen-1 satellite (GF-1) was launched in China on 26 April 2013. It carries two 2 m resolution panchromatic/8 m resolution multispectral cameras and four 16 m resolution wide-field-view (WFV) multispectral cameras, with a combined swath of 800 km and a swing angle of 35°. The GF-1 WFV images (GF-1 images) contain four bands with respective spectral ranges of 0.45–0.52 μm, 0.52–0.59 μm, 0.63–0.69 μm, and 0.77–0.89 μm. The Gaofen-6 satellite (GF-6) was launched in China on 2 June 2018. It carries a 2 m panchromatic/8 m multispectral high-resolution camera and a 16 m multispectral medium-resolution wide-field-view (WFV) camera with a swath of 800 km. The GF-6 WFV images (GF-6 images) consist of eight bands, with the first four bands having the same spectral range as the GF-1

images and the remaining four bands having spectral ranges of 0.69–0.73 μm, 0.73–0.77 μm, 0.40–0.45 μm, and 0.59–0.63 μm, respectively. At present, a large number of GF-1 and GF-6 images have been obtained and are widely used in many industries. For example, on 7 February 2021, a catastrophic mass flow descended in Chamoli, Uttarakhand, India, with over 200 people killed or missing [1]. In similar disasters, GF-1 and GF-6 images before and after the disaster can be used to quickly analyze and evaluate the damage. China has made GF-1 and GF-6 images freely available to the entire world.

The rational function model (RFM) can achieve high accuracy in fitting various physical sensor models, and its interpolation calculation is stable and accurate. In addition, the rational function model has almost the same accuracy as the physical sensor model [2,3]. Space Imaging was first used to employ the RFM for IKONOS satellite images; the Open Geospatial Consortium (OGC) has adopted rational polynomial coefficients (RPCs) as an incidental image parameter standard since 1999 [4]. It is now widely used to produce various high-resolution satellite images, and all public L1-level images include RPC files.

GF-1/GF-6 images also have their own RPC files. For GF-1/GF-6 images, can the accuracy of RFM meet practical requirements?

At present, some of the literature has studied the positioning accuracy of GF-6 images. Yin et al. (2023) [5] used an error-compensated RPCs model for the accurate orientation of GF-6 WFV stereo images, and a precise RPC model was obtained using the traditional beam adjustment method. However, the geometric accuracy evaluation results only show the relative accuracy between the two GF-6 images, not the absolute geometric accuracy of the GF-6 images. In other research [6], the direct geometric accuracy and the geometric accuracy with different adjustment models were evaluated. The direct geometric accuracy is approximately the distance of one ground sample, and the image geometric accuracy is 0.5–1.0 pixels. However, this paper only briefly describes the evaluation method, and the accuracy of a small number of GF-6 images with high direct geometric accuracy was evaluated. Wang et al. (2023) [7] proposed a sensor correction method based on virtual CMOS with distortion for the GF-6 WFV camera, which can improve the internal relative accuracy and the registration accuracy. However, the camera parameters are hard to obtain.

Therefore, in order to more comprehensively evaluate the geometric accuracy of GF-1 and GF-6 images, a geometric accuracy evaluation method is proposed in this paper to address the issue. The proposed method begins with RPC correction, and an RFM and a refined RFM are constructed and used to complete RPC correction. The RFM is refined via affine transformation based on control points (CPs), which results in a refined RFM. Then, an automatic matching method is proposed to generate a large number of uniformly distributed control points (CPs). Finally, these CPs are used to complete the geometric accuracy evaluation of the RFM and refined RFM of GF-1/GF-6 images.

A crucial step in evaluating the geometric accuracy of the RFM is obtaining many uniformly distributed CPs using an automatic matching method with high accuracy and speed.

Automatic matching based on point features has been widely used in remote sensing image matching, from the early Forstner [8], Harris [9], and SUSAN [10] methods to the SIFT [11], SURF [12], and other improved methods with rotation and scale invariance. After extracting feature points, automatic feature matching must be completed. Automatic feature matching primarily focuses on how feature points are described and how their corresponding similarity is measured. Typically, the description of feature points and similarity measures is based on the following descriptors: (1) A variety of similarity measures based on templates, such as the correlation coefficient, phase correlation, and mutual information. (2) Descriptors of various invariant features can form feature vectors, and the similarity measures for the feature vectors are primarily Euclidean distance or other distances. Methods such as SIFT and SURF have feature descriptions with better rotation and scale invariance, but they are computationally slow. Therefore, faster feature matching methods, such as BRIEF [13] and ORB [14], have been proposed.

SIFT and improved methods are widely used for remote sensing image matching. Due to the large size of a single GF-1/GF-6 image and the medium spatial resolution

of GF-1/GF-6 images, SIFT and improved methods have the following problems: high hardware resource consumption, few or no CPs in areas with inconspicuous features, and a slow computation speed. Due to the use of more grayscale information, template matching based on normalized correlation coefficients has a high success rate and strong stability for medium-spatial-resolution remote sensing images with minimal rotation and scale differences.

On the basis of research on existing methods, an automatic matching method based on Harris, BRIEF, and template matching is proposed, which can generate a large number of uniformly distributed and high-accuracy CPs.

Google Earth images have been orthorectified; the research results on the geometric accuracy of Google Earth images (Google images) [15–17] indicate that the average geometric accuracy of Google Earth images is greater than 3 m. Therefore, the 14th-layer Google images of the corresponding area are used as reference images for the geometric accuracy evaluation of GF-1/GF-6 images.

The experimental results showed that the proposed method is superior to SIFT and BRIEF in terms of the number, distribution, and accuracy of the CPs, and the processing speed of the proposed method is much better than that of SIFT.

The accuracy of the RFM and refined RFM was evaluated separately in the experiments. In order to evaluate the geometric accuracy of the RFM, 10,561 GF-1 and GF-6 images of China, Brazil, and France were obtained, their geometric accuracy was statistically analyzed according to the different countries, and the local geometric distortions of the GF-1/GF-6 images were analyzed using eight GF-1 images and eight GF-6 images. Then, the geometric accuracy of the refined RFM was evaluated using the four GF-1 images and four GF-6 images.

The geometric accuracy evaluation results in this paper can not only help researchers to understand the geometric accuracy of China's GF1/GF6 images, but also help researchers to develop high-precision geometric processing algorithms of GF1/GF6 images.

## 2. Geometric Accuracy Evaluation Method

### 2.1. Workflow of Proposed Method

A geometric accuracy evaluation method is proposed in this paper to evaluate the accuracy of the RFM. The proposed method is divided into three major steps: (1) the RPC correction of GF-1/GF-6 images was completed using the RFM and refined RFM, respectively; (2) the automatic matching of RPC-corrected image and Google image of the corresponding area was completed to obtain a large number of CPs; and (3) the geometric accuracy of the whole image was computed using these CPs and the local geometric distortions of the single image were analyzed, as shown in Figure 1.

1. RPC correction

The RFM was constructed using the RPCs and Shuttle Radar Topography Mission (SRTM) 90 m DEM. The RFM is usually refined via affine transformation based on some CPs [2]. This is defined as follows:

$$\begin{aligned}
\Delta l = l' - l = a_0 + a_l \cdot l + a_s \cdot s \\
\Delta s = s' - s = b_0 + b_l \cdot l + b_s \cdot s
\end{aligned} \tag{1}$$

where $(\Delta l, \Delta s)$ express the discrepancies between the measured line and sample coordinates $(l', s')$ and the RFM projected image coordinates $(l, s)$, and the coefficients $a_0, a_l, a_s, b_0, b_l,$ and $b_s$ are the adjustment parameters for each image.

In order to solve the coefficients of affine transformation, at least three pairs of CPs are required. In this paper, CPs were manually selected from GF-1/GF-6 images and reference images.

Then, the RPC correction of the GF-1/GF-6 images was completed using the RFM and refined RFM, respectively. The single GF-6 image was separated into three images for storage purposes, so the geometric accuracy was evaluated separately.

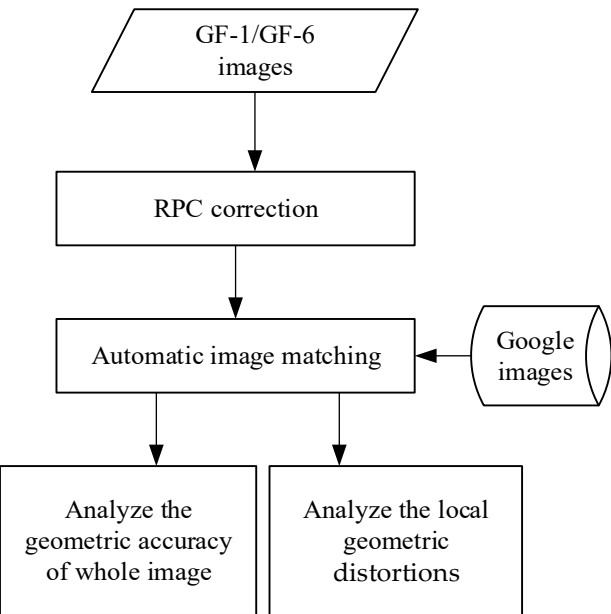

**Figure 1.** Workflow of proposed method.

2.  Automatic image matching

In this paper, an automatic matching method based on Harris, BRIEF, and template matching is proposed to complete automatic matching between an RPC-corrected GF-1/GF-6 image and a Google image of the corresponding area, which can generate a large number of uniformly distributed and high-accuracy CPs, as detailed in the description in Section 2.2.

3.  Geometric accuracy analysis

The root mean square error (*RMSE*) is a measure of the normalized distance between the observed and the predicted data [18,19]. Here, the root mean square error (*RMSE*) was calculated as the geometric accuracy of the whole image using all CPs obtained from automatic matching, as shown below:

$$RMSE = \sqrt{\frac{1}{n}\sum_{i=1}^{n}\left[(x_i - x_i')^2 + (y_i - y_i')^2\right]} \bigg/ d_s \tag{2}$$

For each control point (CP) on the GF-1/GF-6 image, the geometric error value (GEV) was calculated as shown below:

$$\sqrt{(x_i - x_i')^2 + (y_i - y_i')^2} \bigg/ d_s \tag{3}$$

In Equations (2) and (3), *n* represents the total number of CPs, $(x_i, y_i)$ represents the geographic coordinates of the CP on the GF-1/GF-6 image, $(x_i', y_i')$) represents the geographic coordinates of the CP on the reference image, and $d_s$ represents the resolution of the GF-1/GF-6 image.

For the GF-1/GF-6 images corrected by the RFM, the geometric accuracy of the whole image and the local geometric distortions were analyzed: (1) For many GF-1 and GF-6 images, histograms of the RMSE values were statistically analyzed according to different countries, the results of which were used to evaluate the geometric accuracy of whole image. (2) The local geometric distortions can affect the accuracy of the refined RFM; therefore, eight GF-1 images and eight GF-6 images from different countries and with different geometric accuracies were selected to evaluate the local geometric distortions. For

each image, the geometric error value of each CP was calculated; then, a plot of geometric errors and a histogram of geometric error values were obtained. Lastly, the local geometric distortions for each image were analyzed using a plot of geometric errors and a histogram of geometric error values.

For the GF-1/GF-6 images corrected by the refined RFM, the RMSE values were calculated and analyzed.

### 2.2. Automatic Matching Method

This paper aims to evaluate the geometric accuracy of the RFM using the CPs obtained via automatic matching for a large number of GF-1 and GF-6 images. In order to obtain more accurate evaluation results, the CPs obtained via the automatic matching method had to be distributed as evenly as possible. In addition, since many images were completed in this paper, the processing speed of the automatic matching method had to be as fast as possible.

An automatic matching method based on Harris, BRIEF, and template matching was proposed in this paper. Harris is an extraction algorithm for feature points. BRIEF uses binary strings as an efficient feature point descriptor. BRIEF is faster than traditional descriptors such as SURF and SIFT in terms of speed. Template matching has a high success rate and good stability for the automatic matching of medium-spatial-resolution remote sensing images with small differences in scale and rotation.

Automatic image matching was completed based on image chunking, with the following steps: Firstly, the reference image was reprojected to the same projection and resolution as the RPC-corrected GF-1/GF-6 image, which can reduce the scale difference between the reference image and the RPC-corrected GF-1/GF-6 image. Secondly, the feature points were extracted from the original image block and the reference image block using Harris, and then, BRIEF was used to match the feature points. Lastly, for each feature point that failed to be matched, template matching was used. After all the feature points were matched, the RANSAC approach was used to detect incorrect CPs, and then, control point homogenization was carried out, as shown in Figure 2.

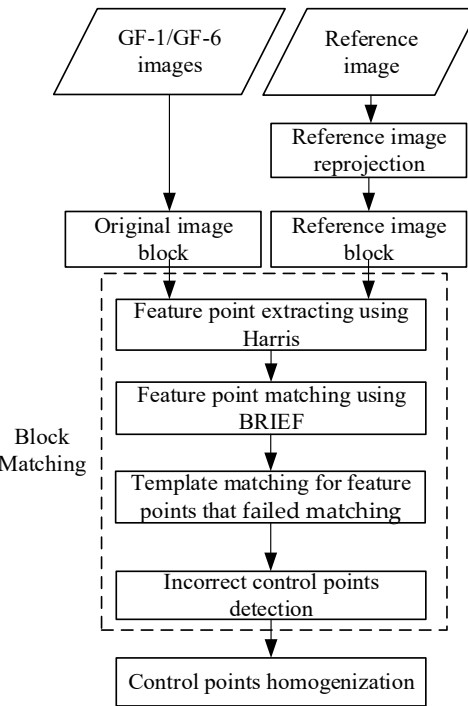

**Figure 2.** Workflow of automatic matching method.

The following section describes the primary technologies involved in the automatic matching method proposed in this paper:

1. Image Chunking

The GF-1/GF-6 images were evenly chunked according to a size of B × B to form several original image blocks. For each original image block, the geographic coordinates of the center point of the original image block were used to calculate the center point of the reference image block in the corresponding position. The corresponding reference image block was extracted using the calculated center point. The size of the reference image block was bigger than that of the original image block. The size of B affects the memory footprint and processing speed of the algorithm. The larger the B value, the higher the memory footprint and the faster the processing. Therefore, it was necessary to determine the size of B according to the algorithm's memory footprint, the size of the computer memory, and the number of parallel processing images.

2. Feature point extraction using Harris

Harris is used to detect feature points in images and videos. Harris finds significant changes in image gradients for a local area (window) and is often used to extract feature points from images. In this paper, Harris was used to extract feature points from each image block.

3. Feature point matching using BRIEF

For each feature point on the original image block and the reference image block, BRIEF was used to separately generate a 512-bit binary feature string. Then, the Hamming distance was used to complete feature string matching.

4. Template matching

For each feature point that failed to be matched, a template window was extracted with the feature point at the center. Then, the corresponding position of the feature point on the reference image block was calculated based on the same geographic coordinate. The search window (where the size was larger than the template window) was extracted with this position at the center, and then, template matching was completed using Fast NCC [20].

5. Incorrect control point detection

Although the proposed method can obtain many accurate CPs, there are still some incorrect CPs. Therefore, it was necessary to remove incorrect CPs. The RANSAC [21] method is usually used to detect incorrect CPs and can obtain good results in many cases. Therefore, the RANSAC method was used to detect incorrect CPs for each image block.

6. Control point homogenization.

Firstly, the original image was gridded according to a size of 400 × 400; all CPs were assigned to various grids based on the control point coordinates of the original image. Then, only the CP with the optimal matching degree was maintained for each grid. The rules used to determine the optimal matching degree were as follows: (1) if all CPs in one grid are obtained from BRIEF, the CP with the smallest Hamming distance is maintained; (2) if all CPs in one grid are obtained from template matching, the CP with the maximum NCC value is maintained; (3) if CPs in one grid are obtained from BRIEF and template matching, the CP with the smallest Hamming distance is maintained.

## 3. Experiments for Geometric Accuracy Evaluation Method

### 3.1. Experimental Data

In this experiment, the advantages and disadvantages of BRIEF, SIFT, and the proposed method are compared from the perspectives of the number of CPs, the distribution of CPs, the accuracy of the CPs, and processing time. The RPC correction of the GF-1/GF-6 images

was completed using the RFM. The experimental images were four GF-1 images; the details are shown in Table 1. In this and subsequent experiments, the value of B was 5000.

**Table 1.** Experimental GF-1 image information for geometric accuracy evaluation method.

| Number | Sensors | Imaging Time | Country |
|---|---|---|---|
| No. 1 | GF-1 WFV1 | 21 February 2018 | China |
| No. 2 | GF-1 WFV1 | 23 December 2018 | China |
| No. 3 | GF-1 WFV1 | 17 March 2014 | China |
| No. 4 | GF-1 WFV2 | 23 April 2021 | France |

*3.2. Experimental Results Regarding Number of CPs*

For all experimental images, the proposed method obtains more CPs than SFIT and BRIEF. The number of CPs obtained by BRIEF is less than that of SIFT, as illustrated in Figure 3.

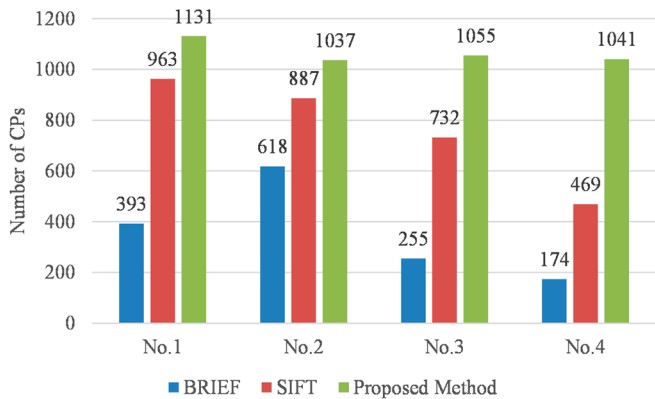

**Figure 3.** Histogram of number of CPs for different methods.

*3.3. Experimental Results Regarding Distribution of CPs*

Figures 4–7 show the distribution of CPs for different methods. For all images, the distribution of CPs obtained from the proposed method is relatively uniform. For all images, the distribution obtained from the BRIEF method is not uniform, and some places do not have CPs. The distributions of CPs obtained from SIFT in the No. 1 image and No. 2 image are generally uniform. The distributions of CPs obtained from SIFT in the No. 3 image and No. 4 image are not uniform.

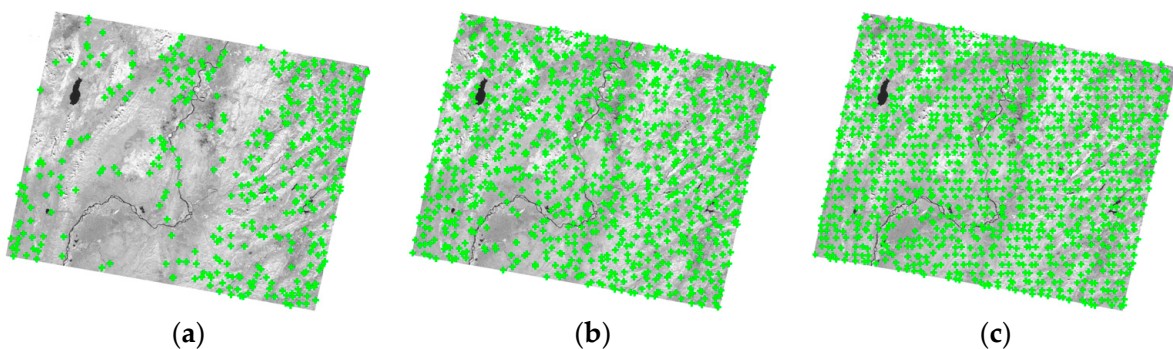

(**a**)  (**b**)  (**c**)

**Figure 4.** The distribution of CPs of different methods for No. 1 image. (**a**) BRIEF. (**b**) SIFT. (**c**) Proposed method.

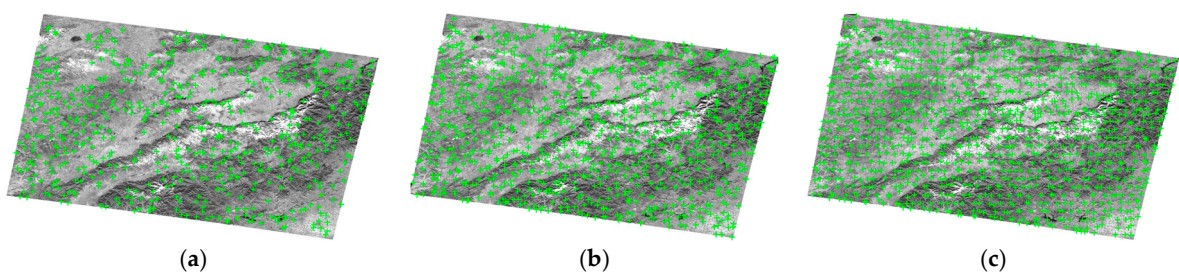

(**a**)  (**b**)  (**c**)

**Figure 5.** The distribution of CPs of different methods for No. 2 image. (**a**) BRIEF. (**b**) SIFT. (**c**) Proposed method.

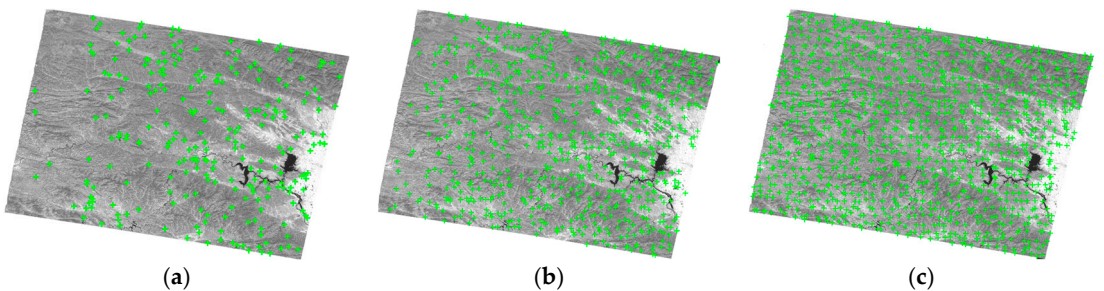

(**a**)  (**b**)  (**c**)

**Figure 6.** The distribution of CPs of different methods for No. 3 image. (**a**) BRIEF. (**b**) SIFT. (**c**) Proposed method.

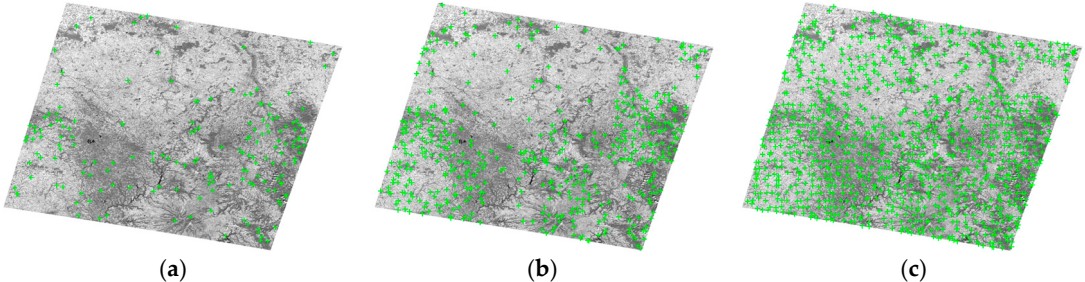

(**a**)  (**b**)  (**c**)

**Figure 7.** The distribution of CPs of different methods for No. 4 image. (**a**) BRIEF. (**b**) SIFT. (**c**) Proposed method.

### 3.4. Experimental Results Regarding Processing Time

The processing times of the different methods are presented in ascending order: BRIEF, the proposed method, and SIFT. The processing time of SIFT is much higher than that of BRIEF and the proposed method, as illustrated in Figure 8.

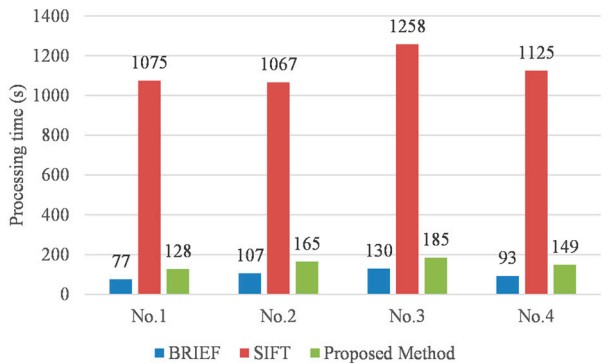

**Figure 8.** Histogram of processing times for different methods.

### 3.5. Experimental Results Regarding Evaluation Accuracy

In order to verify the accuracy of the automatic geometric evaluation of BRIEF, SIFT, and the proposed method, a manual method was used to obtain an accurate RMSE. The steps of the manual method were as follows: first, about 60 uniformly distributed CPs were manually selected on the RPC-corrected image and reference image; then, these CPs were used to calculate the RMSE of each image. The manually selected CPs have high accuracy and a uniform distribution, so the RMSE obtained from the manual method was used to evaluate the accuracy of BRIEF, SIFT, and the proposed method.

The RMSE values obtained from the different methods used for the four experimental images are illustrated in Figure 9. For the No. 1 image, the RMSE value of SIFT is equal to that of the manual method, and the RSME values of BRIEF and the proposed method differ very little from that of the manual method. For the No. 2 image and No. 4 image, the RMSE value of the proposed method is equal to that of the manual method. For the No. 3 image, the RMSE of the proposed method is closest to that of the manual method.

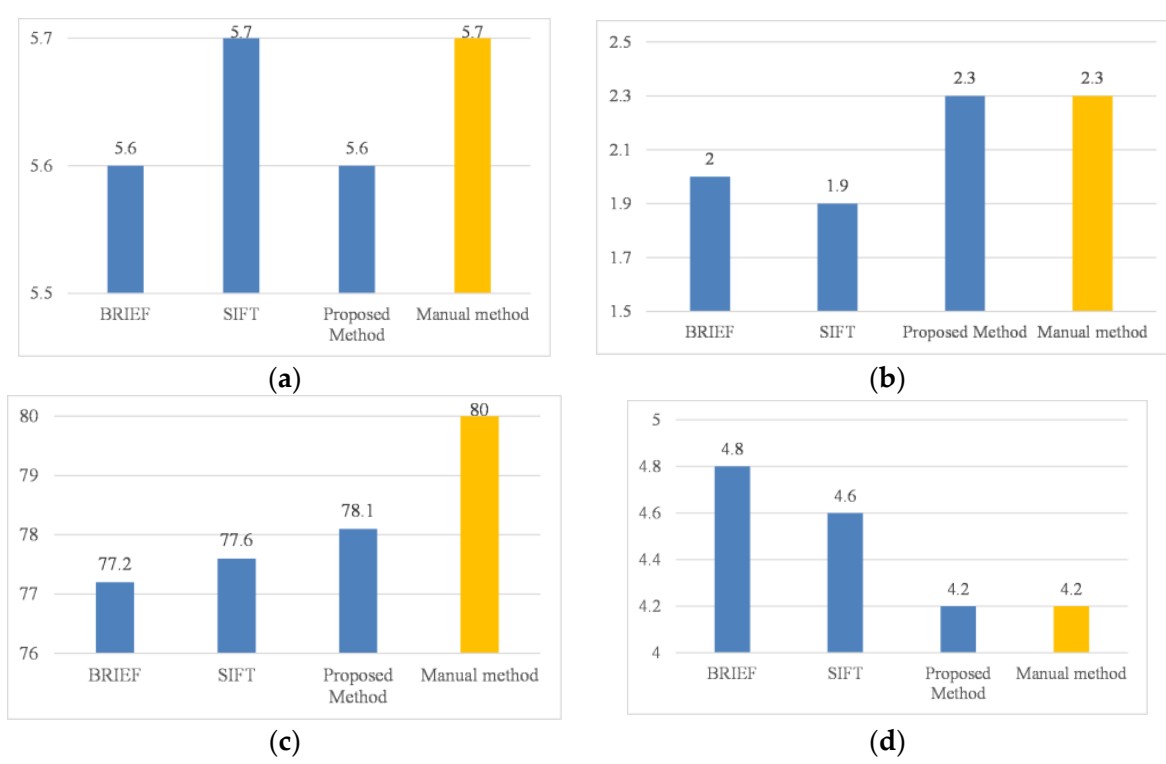

**Figure 9.** The RMSE values obtained from the different methods used for the four experimental images. (**a**) No. 1 image. (**b**) No. 2 image. (**c**) No. 3 image. (**d**) No. 4 image.

According to a comprehensive comparison of the number of CPs, the distribution of CPs, the accuracy of the CPs, and processing time, the proposed method is superior to SIFT and BRIEF in terms of the geometric accuracy evaluation of GF1/GF6 images.

## 4. Geometric Accuracy Evaluation Results
### 4.1. The Results for the RFM
#### 4.1.1. Experimental Data

The experimental images consist of 10,561 GF-1 and GF-6 images of China, Brazil, and France, including 5539 GF-1 images and 5022 GF-6 images, with GF-1 images imaged from 2013 to 2021 and GF-6 images imaged from 2018 to 2021. The details of the experimental images are shown in Table 2.

**Table 2.** Experimental GF-1/GF-6 image information for geometric accuracy evaluation.

| Country | Sensor | Number of Images | Imaging Time Range |
|---------|--------|------------------|--------------------|
| China | GF-1 WFV | 2990 | May 2013–June 2021 |
| | GF-6 WFV | 1077 | November 2018–June 2021 |
| Brazil | GF-1 WFV | 335 | May 2014–September 2020 |
| | GF-6 WFV | 2840 | July 2018–June 2021 |
| France | GF-1 WFV | 2214 | May 2013–June 2021 |
| | GF-6 WFV | 1105 | August 2018–June 2021 |

### 4.1.2. The Geometric Accuracy of the Whole Image

1. Experimental results of GF-1 images

The RMSE values of 2990 GF-1 images of China were statistically analyzed, and the RMSE values ranged from 1 to 37, with 96.5% of the images having RMSE values between 1 and 11; detailed results are shown in Figure 10.

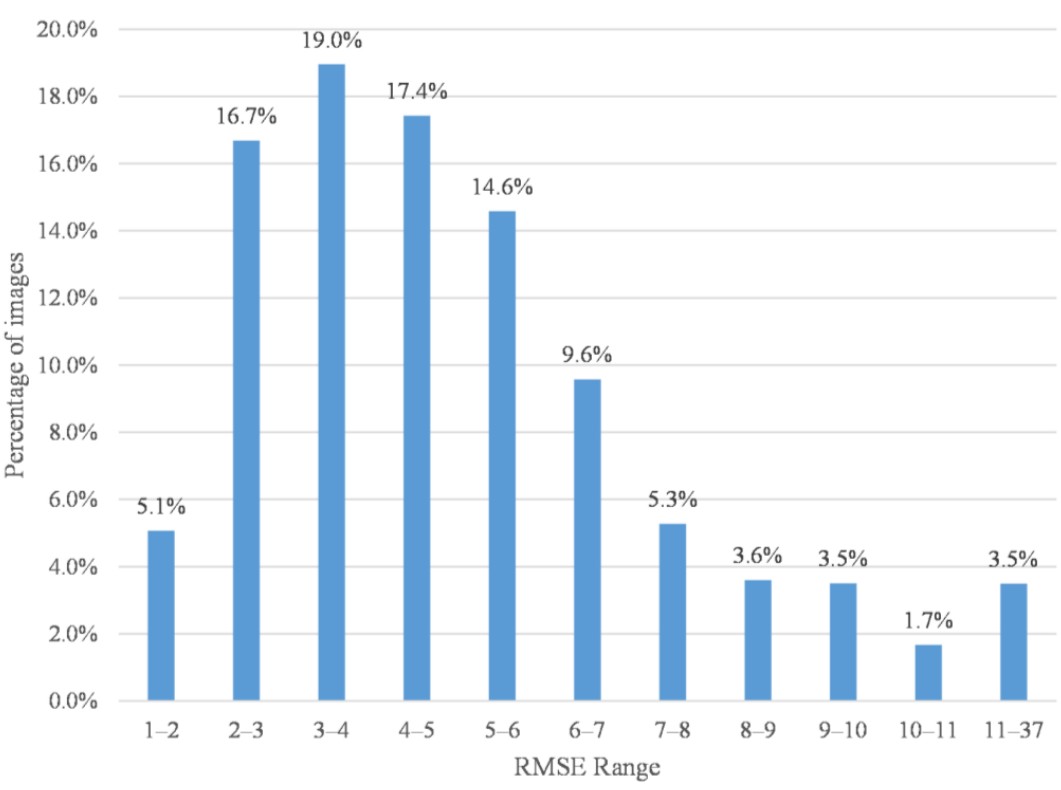

**Figure 10.** RMSE histogram of GF-1 images of China.

The RMSE values of 355 GF-1 images of Brazil were statistically analyzed. The RMSE values ranged from 5 to 22; detailed results are shown in Figure 11.

The RMSE values of 2214 GF-1 images of France were statistically analyzed. The values of RMSE ranged from 2 to 45, with 96.2% images having RMSE values between 2 and 19; detailed results are shown in Figure 12.

By analyzing the RMSE values of GF-1 images of China, France, and Brazil, it was determined that the geometric accuracy of the GF-1 images of China is primarily distributed between 1 and 11, that of the GF-1 images of France is primarily distributed between 2 and 19, and that of the GF-1 images of Brazil is primarily distributed between 5 and 20. The geometric accuracy of the GF-1 images of China is superior to that of France and Brazil. The RMSE values for all GF-1 images cannot meet the practical requirement of being better than 1.0 pixel.

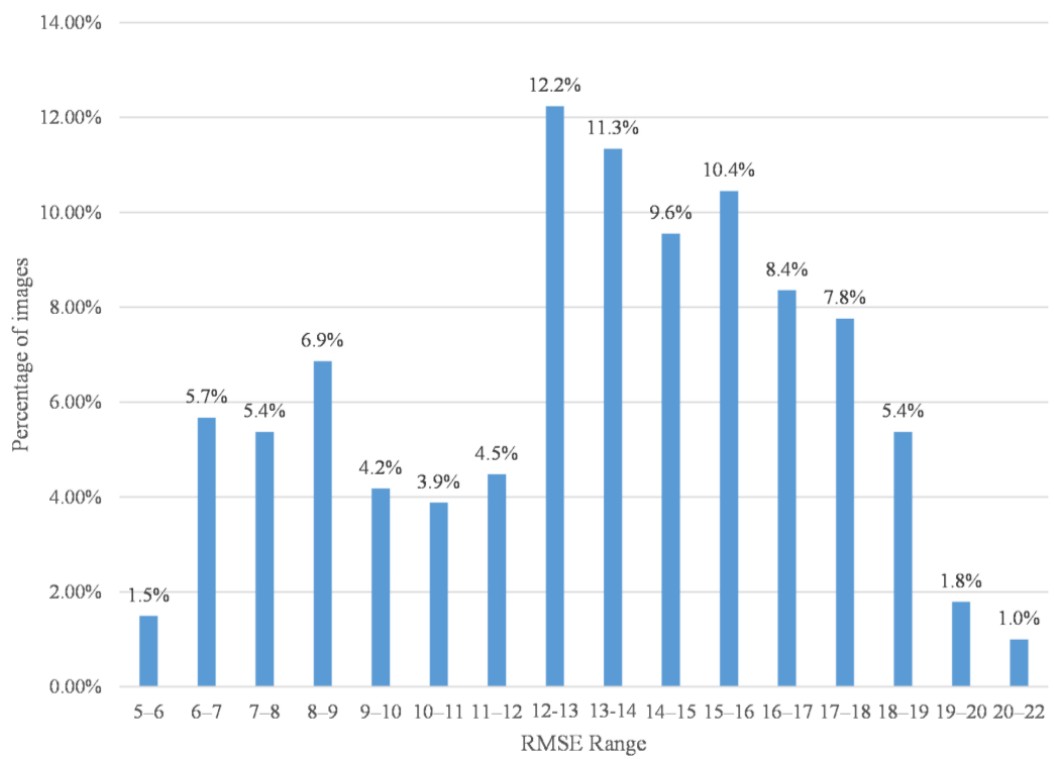

**Figure 11.** RMSE histogram of GF-1 images of Brazil.

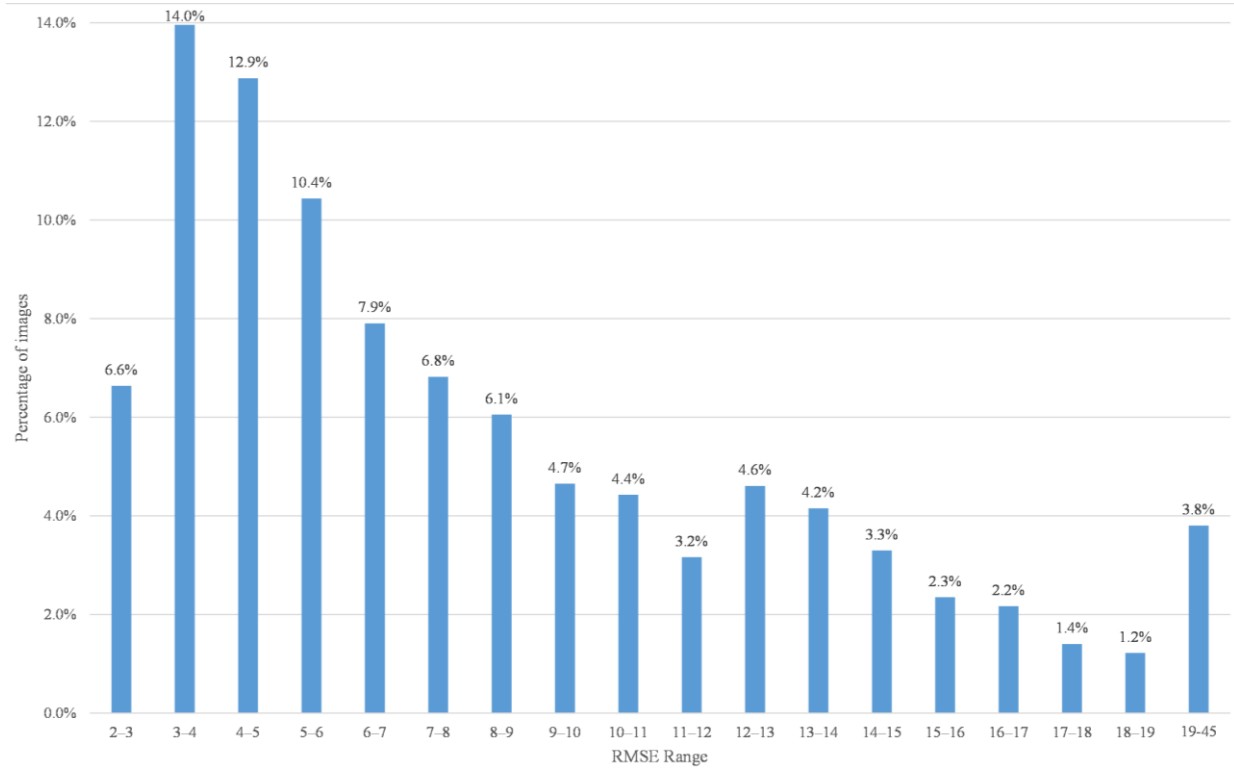

**Figure 12.** RMSE histogram of GF-1 images of France.

2.  Experimental results of GF-6 images

The RMSE values of 1077 GF-6 images of China were statistically analyzed. The RMSE values ranged from 0 to 60, with 99.4% of the images having RMSE values between 0 and 5.0; detailed results are shown in Figure 13.

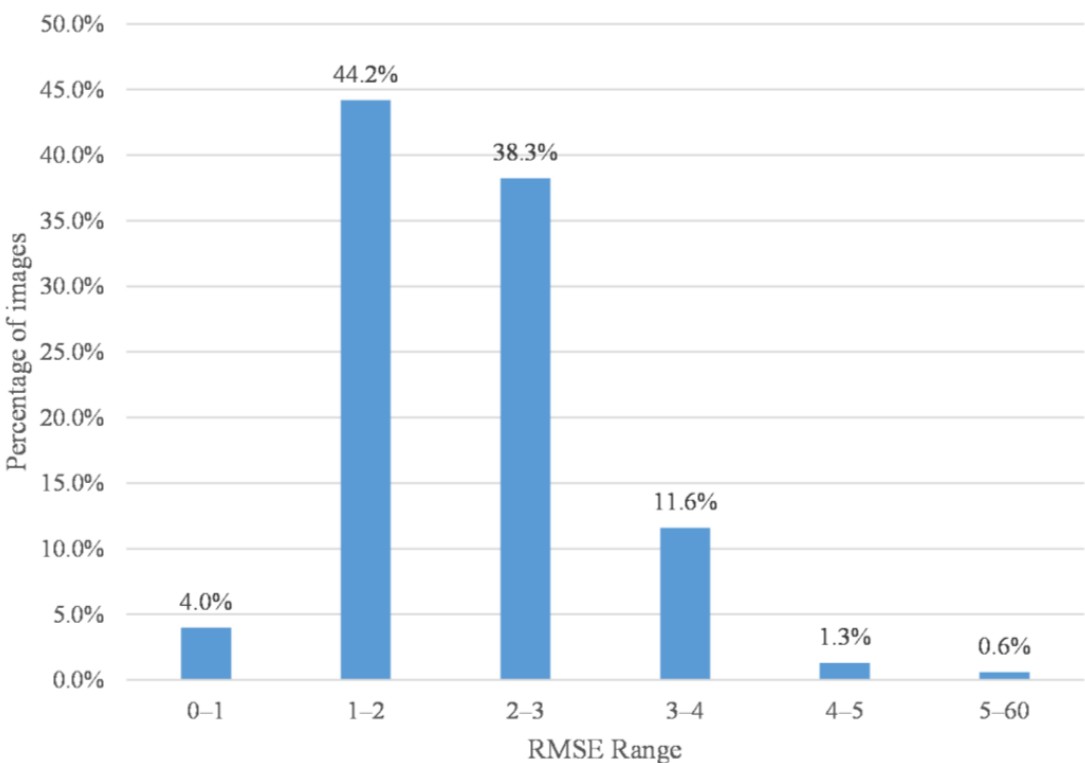

**Figure 13.** RMSE histogram of GF-6 images of China.

The RMSE values of 2840 Brazil GF-6 images were statistically analyzed. The RMSE values ranged from 0 to 17, with 99.1% of the images having values between 0 and 4.0; detailed results are shown in Figure 14.

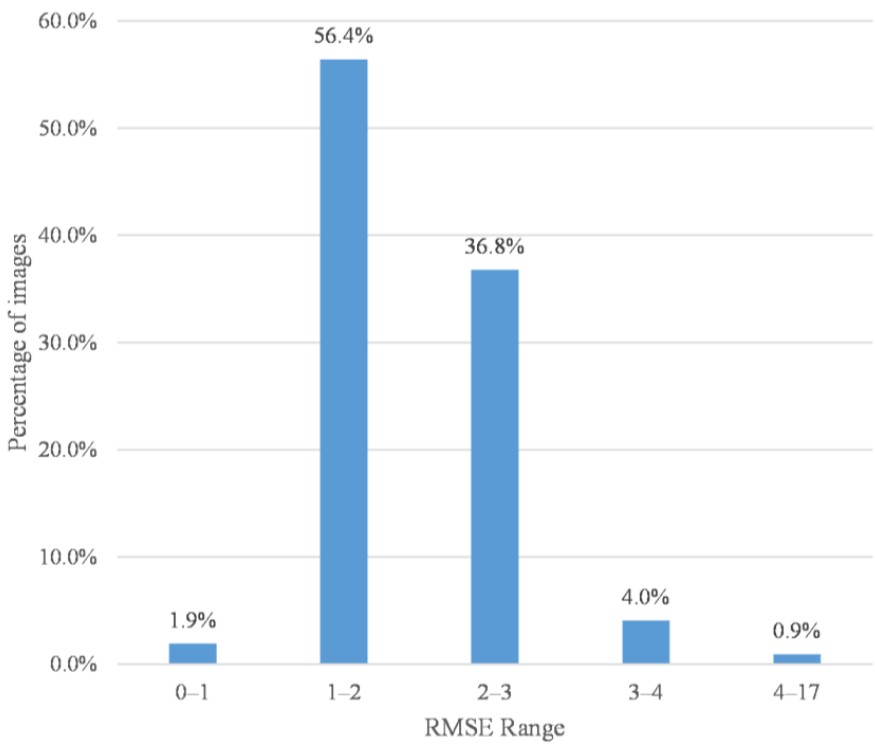

**Figure 14.** RMSE histogram of GF-6 images of Brazil.

The RMSE values of 1105 GF-6 images of France were statistically analyzed. The RMSE values ranged from 0 to 9, with 99.0% of the images having values between 0 and 7; detailed results are shown in Figure 15.

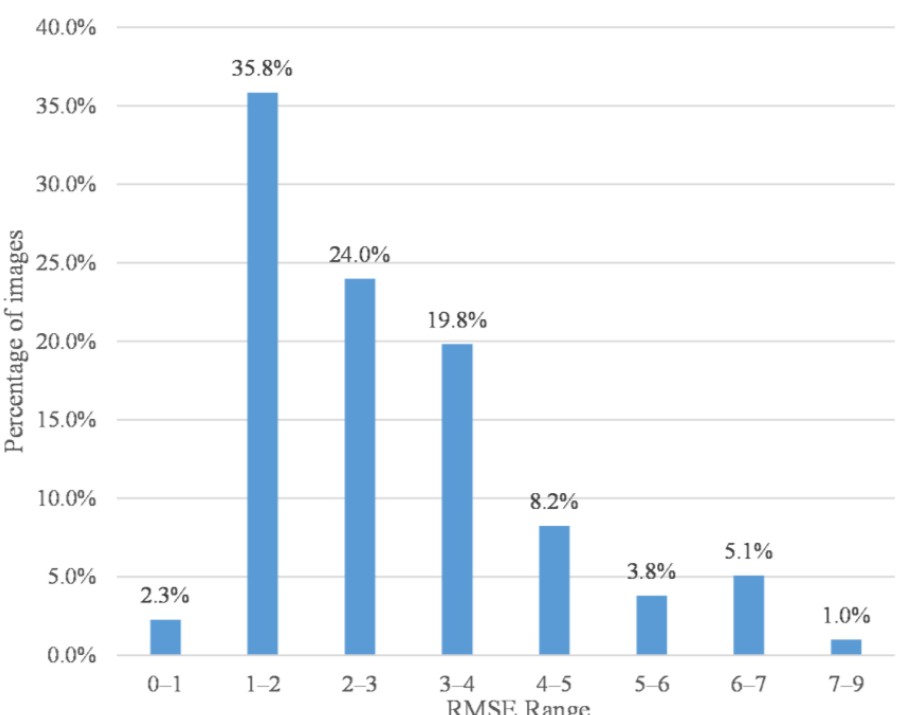

**Figure 15.** RMSE histogram of GF-6 images of France.

The geometric accuracy of the GF-6 images of China is primarily distributed between 0 and 5; the geometric accuracy of the GF-6 images of France is primarily distributed between 0 and 7. The geometric accuracy of the GF-6 images of Brazil is primarily distributed between 0 and 4. There is no significant difference in the geometric accuracy of the three national images, but the RMSE values of a very small number of images can meet the practical requirement of being better than 1.0 pixel.

Comparing the geometric accuracy of GF-6 images and GF-1 images showed that the geometric accuracy of the GF-6 images is significantly superior to that of the GF-1 images. This result shows that the RFM's accuracy for the GF-6 images is better than that for the GF-1 images, and the accuracy of the GF-6 satellite imaging parameters is better than that of GF-1.

The results of the geometric accuracy evaluation of the whole image show that if only the RPCs of the GF-1/GF-6 images are used to build the RFM, the accuracy of most GF-1/GF-6 images cannot meet the practical requirements, and even the RMSE values of some GF-1 images are relatively large, for two main reasons: (1) the satellite imaging parameters are inaccurate, resulting in inaccuracies in the RFM, and (2) when RPC correction is performed, the RFM is not optimized using CPs.

### 4.1.3. Local Geometric Distortions

1.  Experimental results of GF-1 images

Four GF-1 images of China, two GF-1 images of Brazil, and two GF-1 images of France were selected from all the experimental GF-1 images. The local geometric distortions were analyzed using geometric error values and geometric error directions.

The details of the eight experimental images are shown in Table 3.

**Table 3.** Experimental GF-1 image information for analysis of local geometric distortions.

| Number | Sensors | Imaging Time | Country | The Value of RMSE |
|--------|---------|--------------|---------|-------------------|
| No. 1 | GF-1 WFV1 | 23 December 2018 | China | 2.3 |
| No. 2 | GF-1 WFV1 | 17 March 2014 | China | 78 |
| No. 3 | GF-1 WFV1 | 12 January 2021 | China | 3.4 |
| No. 4 | GF-1 WFV3 | 30 August 2019 | China | 7.1 |
| No. 5 | GF-1 WFV4 | 20 July 2017 | Brazil | 11.9 |
| No. 6 | GF-1 WFV3 | 20 July 2017 | Brazil | 10.8 |
| No. 7 | GF-1 WFV1 | 27 April 2021 | France | 10.5 |
| No. 8 | GF-1 WFV2 | 23 April 2021 | France | 4.7 |

The geometric error values of CPs for the No. 1 image range from 0 to 4. There are significant differences in the geometric error directions of CPs, especially the geometric error values and directions in the left area of the image, which is obviously different from other areas. Therefore, the No. 1 image has significant local geometric distortions, as shown in Figure 16.

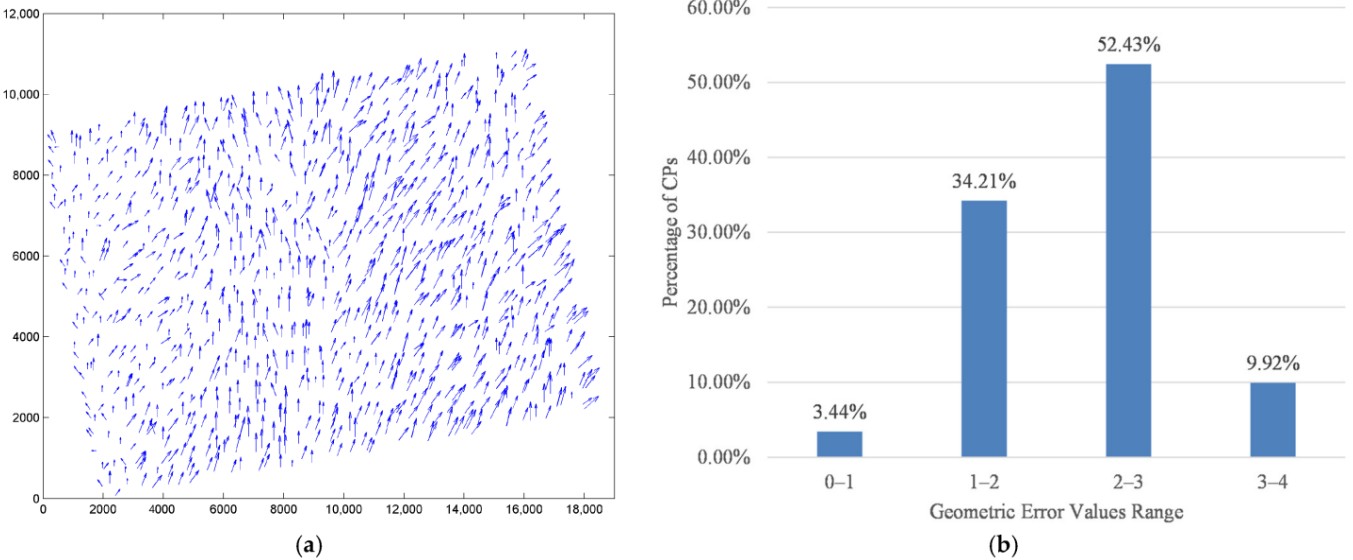

(**a**)     (**b**)

**Figure 16.** Analysis results of local geometric distortions in No. 1 image: (**a**) plot of geometric errors, (**b**) histogram of geometric error values.

The geometric error values of CPs for the No. 2 image range from 64 to 98. There are significant differences in the geometric error values of the CPs, and small differences in the geometric error direction of the CPs. Therefore, the No. 2 image has significant local geometric distortions, as shown in Figure 17.

The geometric error values of the CPs for the No. 3 image range from 1 to 6, and 98.89% of the geometric error values are between 2 and 5. There are small differences in the geometric error values of the CPs, and significant differences in the geometric error directions of the CPs. Therefore, the No. 3 image has significant local geometric distortions, as shown in Figure 18.

The geometric error values of the CPs for the No. 4 image range from 3 to 8. There are significant differences in the geometric error directions of the CPs, especially the error directions on the right side of the image, which are significantly different from the other parts. Therefore, the No. 4 image has significant local geometric distortions, as shown in Figure 19.

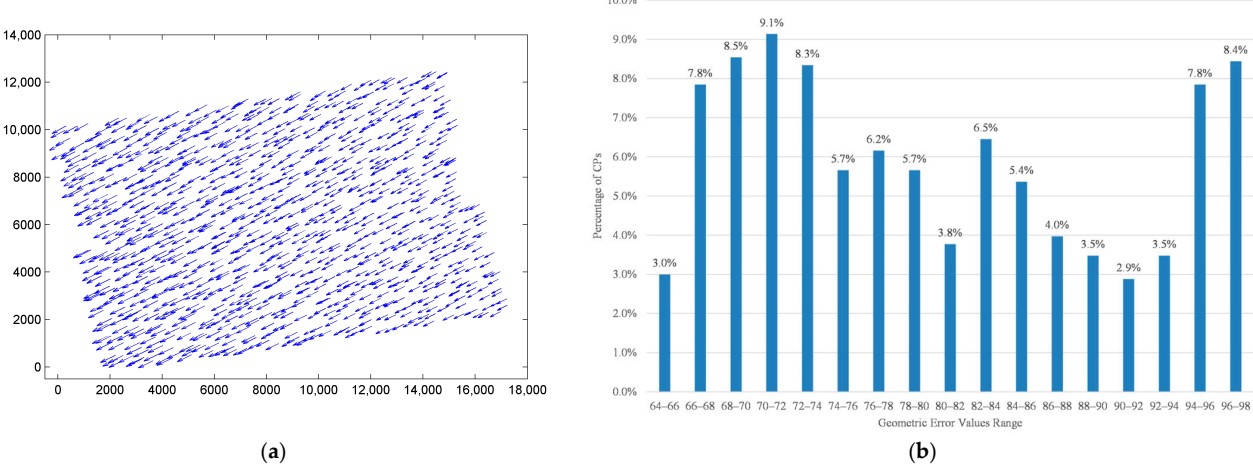

**Figure 17.** Analysis results of local geometric distortions in No. 2 image: (**a**) plot of geometric errors, (**b**) histogram of geometric error values.

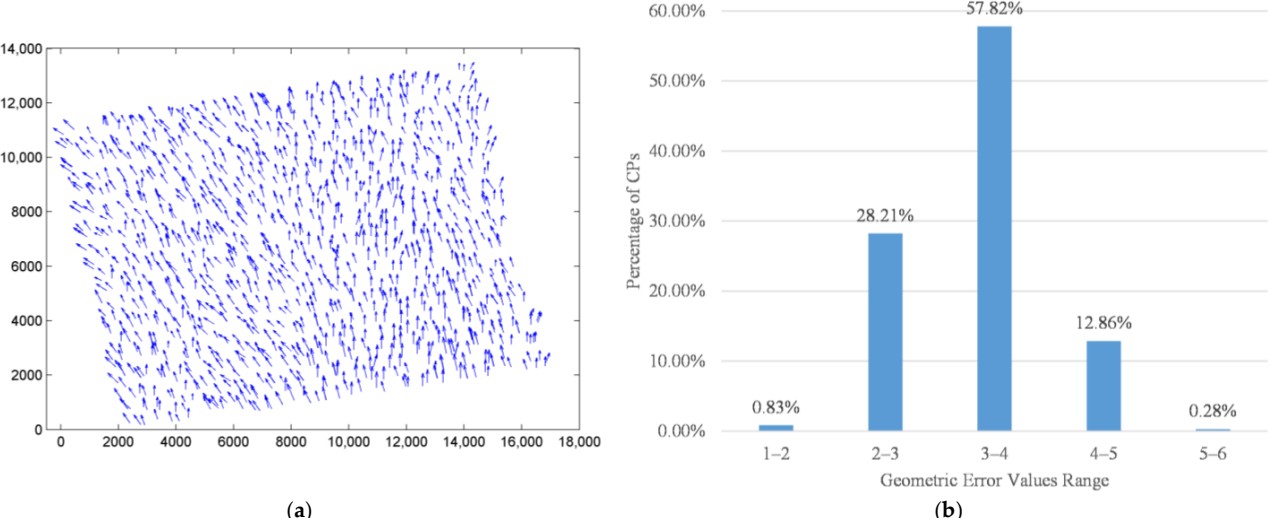

**Figure 18.** Analysis results of local geometric distortions in No. 3 image: (**a**) plot of geometric errors, (**b**) histogram of geometric error values.

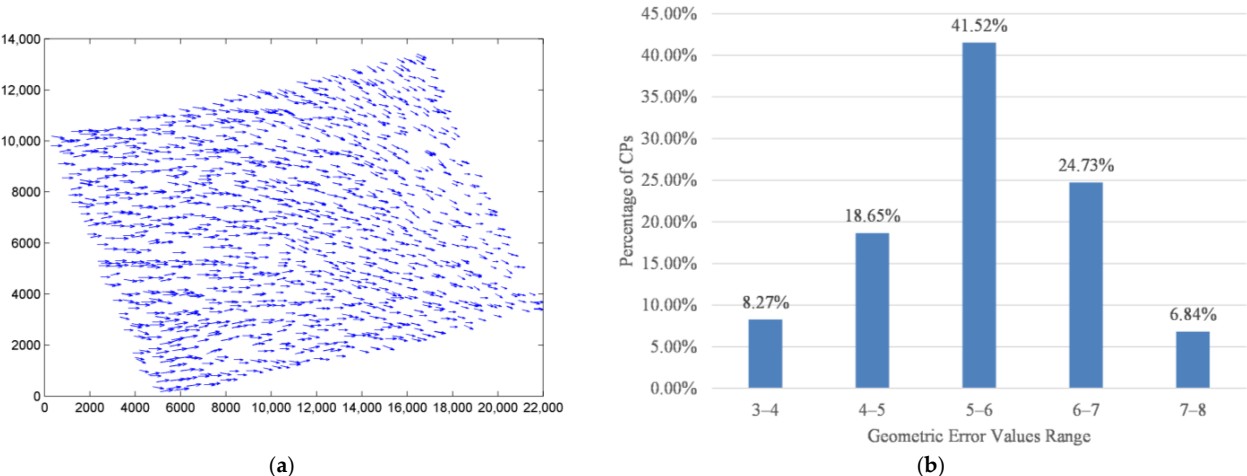

**Figure 19.** Analysis results of local geometric distortions in No. 4 image: (**a**) plot of geometric errors, (**b**) histogram of geometric error values.

The geometric error values of the CPs for the No. 5 image range from 9 to 13. There are small differences in the geometric error values and directions of the CPs. Therefore, the No. 5 image has small local geometric distortions, as shown in Figure 20.

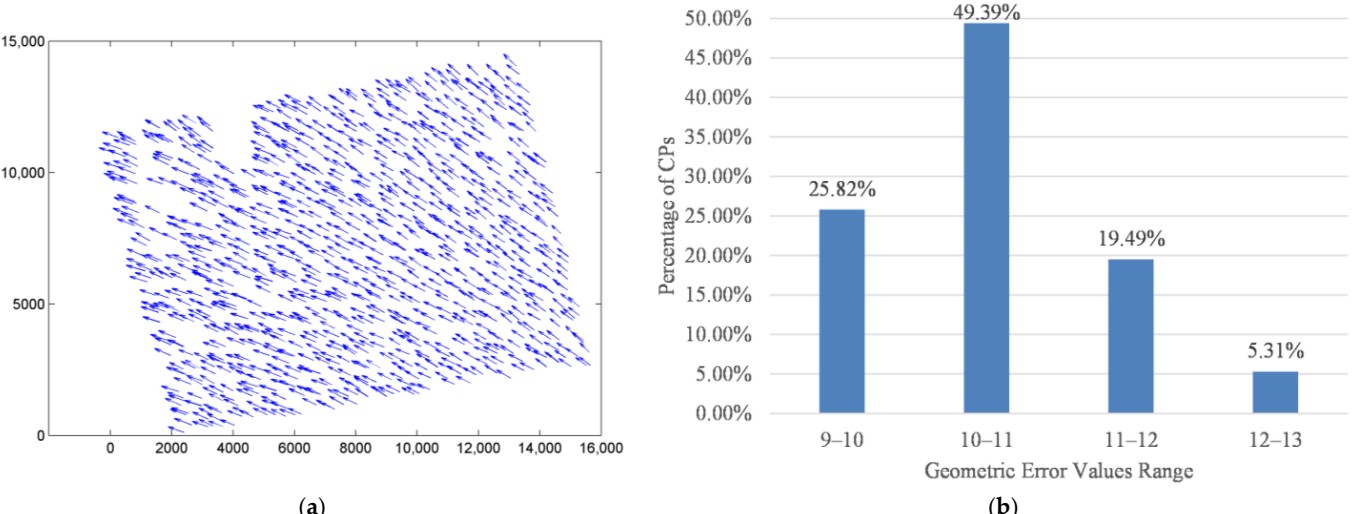

(**a**)  (**b**)

**Figure 20.** Analysis results of local geometric distortions in No. 5 image: (**a**) plot of geometric errors, (**b**) histogram of geometric error values.

The geometric error values of the CPs for the No. 6 image range from 8 to 13. There are small differences in the geometric error values and directions of the CPs. Therefore, the No. 6 image has small local geometric distortions, as shown in Figure 21.

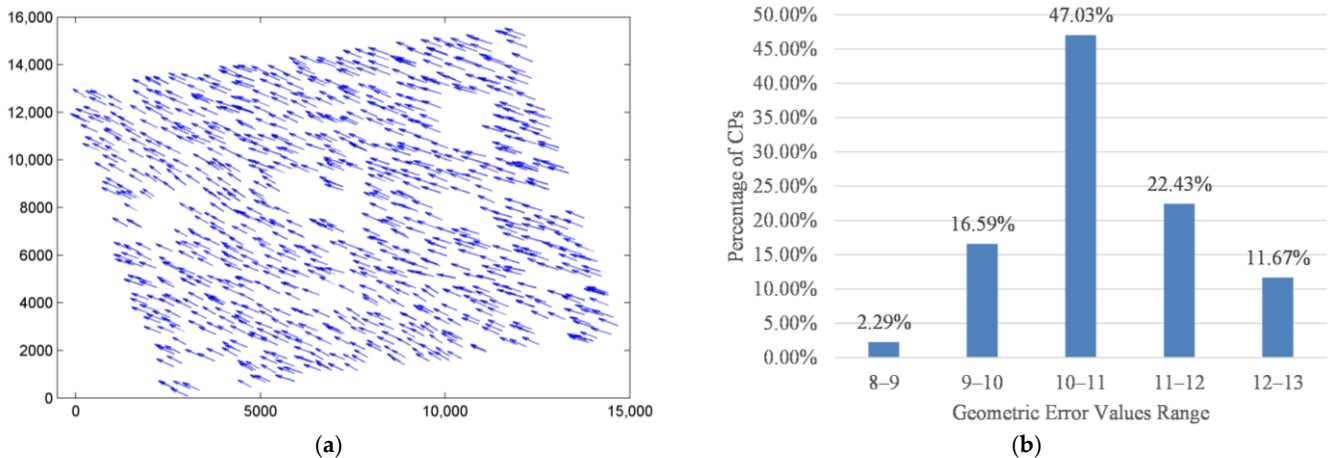

(**a**)  (**b**)

**Figure 21.** Analysis results of local geometric distortions in No. 6 image: (**a**) plot of geometric errors, (**b**) histogram of geometric error values.

The geometric error values of the CPs for the No. 7 image range from 1 to 5. There are small differences in the geometric error values of the CPs, and significant differences in the geometric error directions of the CPs. Therefore, the No. 7 image has significant local geometric distortions, as shown in Figure 22.

The geometric error values of the CPs for the No. 8 image range from 2 to 6. There are small differences in the geometric error values of the CPs, and significant differences in the geometric error directions of the CPs, especially in the lower part of the image. Therefore, the No. 8 image has significant local geometric distortions, as shown in Figure 23.

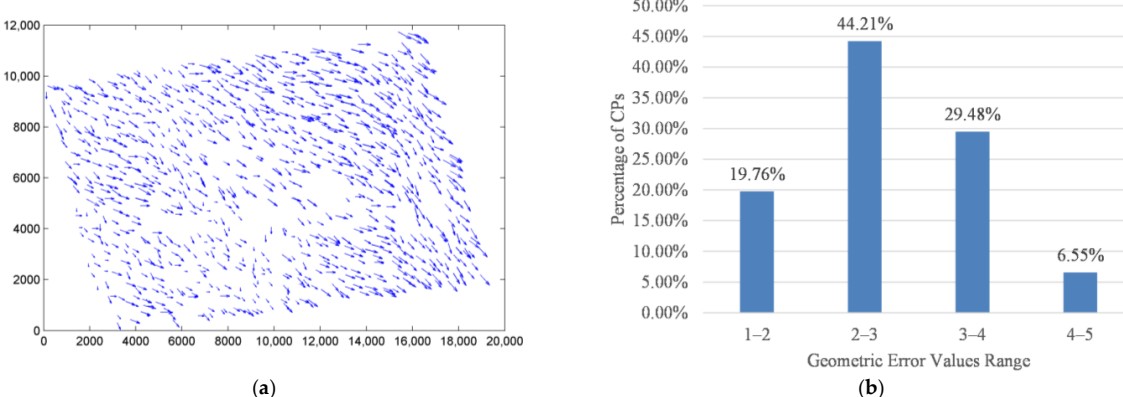

(**a**)                                    (**b**)

**Figure 22.** Analysis results of local geometric distortions in No. 7 image: (**a**) plot of geometric errors, (**b**) histogram of geometric error values.

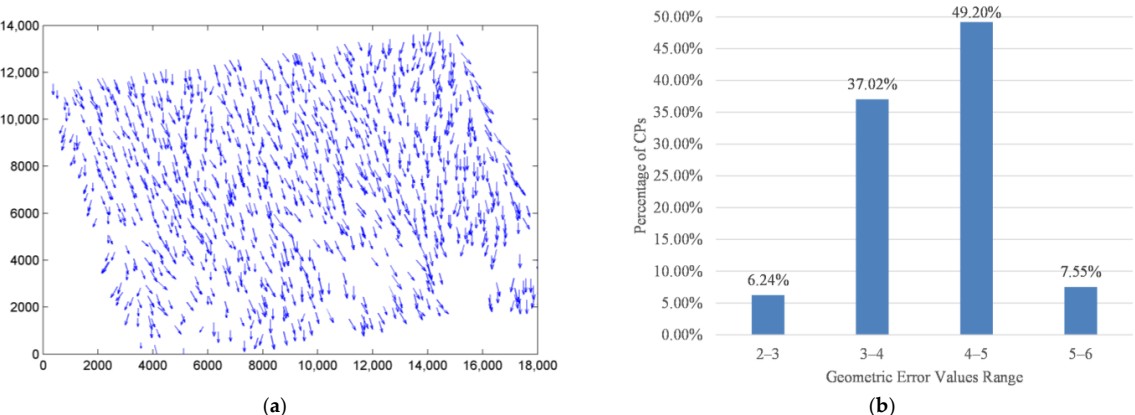

(**a**)                                    (**b**)

**Figure 23.** Analysis results of local geometric distortions in No. 8 image: (**a**) plot of geometric errors, (**b**) histogram of geometric error values.

The results of the analysis of the local geometric distortions for the eight GF-1 images indicate that most of the GF-1 images have significant local geometric distortions, and a few images have small local geometric distortions.

2.    Experimental results of GF-6 images

Four GF-6 images of China, two GF-6 images of Brazil, and two GF-6 images of France were selected from all the experimental GF-6 images. The local geometric distortions were analyzed using the geometric error values and directions.

The details of the eight experimental images are shown in Table 4.

**Table 4.** Experimental GF-6 image information for analysis of local geometric distortions.

| Number | Sensors | Imaging Time | Country | The Value of RMSE |
| --- | --- | --- | --- | --- |
| No. 1 | GF-6 WFV | 26 November 2018 | China | 2.7 |
| No. 2 | GF-6 WFV | 26 January 2019 | China | 1.9 |
| No. 3 | GF-6 WFV | 19 November 2020 | China | 1.9 |
| No. 4 | GF-6 WFV | 14 December 2020 | China | 2.8 |
| No. 5 | GF-6 WFV | 14 September 2020 | Brazil | 16.8 |
| No. 6 | GF-6 WFV | 5 March 2021 | Brazil | 2.2 |
| No. 7 | GF-6 WFV | 15 February 2019 | France | 6.7 |
| No. 8 | GF-6 WFV | 31 March 2021 | France | 1.5 |

The geometric error values of the CPs for the No. 1 image range from 1 to 5; there are significant differences in the geometric error directions of the CPs, but very small differences in the geometric error values of the CPs. Therefore, the No. 1 image has small local geometric distortions, as shown in Figure 24.

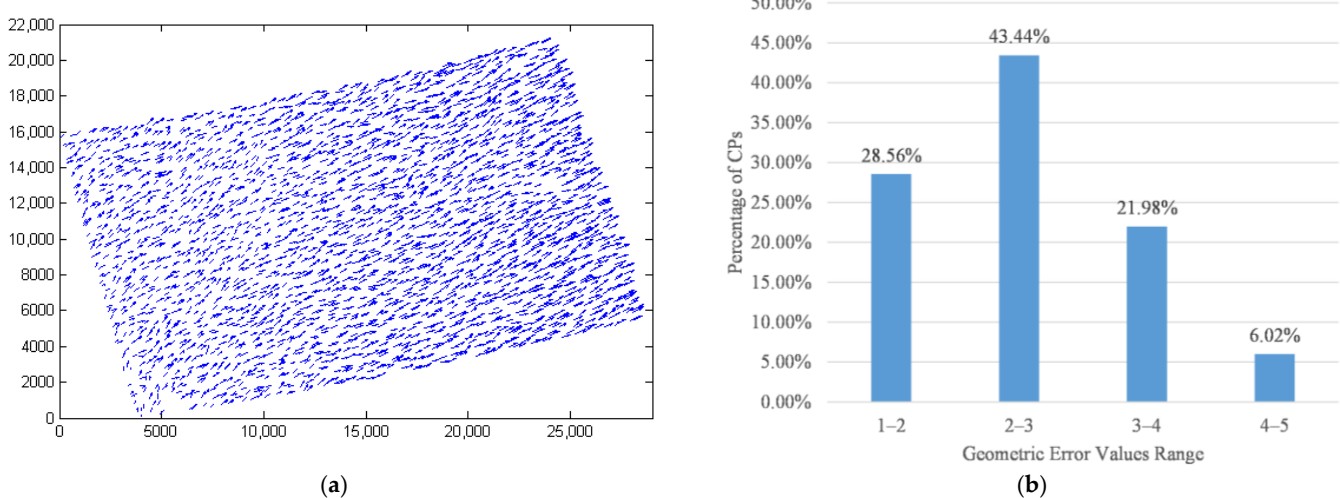

(**a**)                                    (**b**)

**Figure 24.** Analysis results of local geometric distortions in No. 1 image: (**a**) plot of geometric errors, (**b**) histogram of geometric error values.

The geometric error values of the CPs for the No. 2 image range from 0 to 4; there are significant differences in the geometric error directions of the CPs, but very small differences in the geometric error values of the CPs. Therefore, the No. 2 image has small local geometric distortions, as shown in Figure 25.

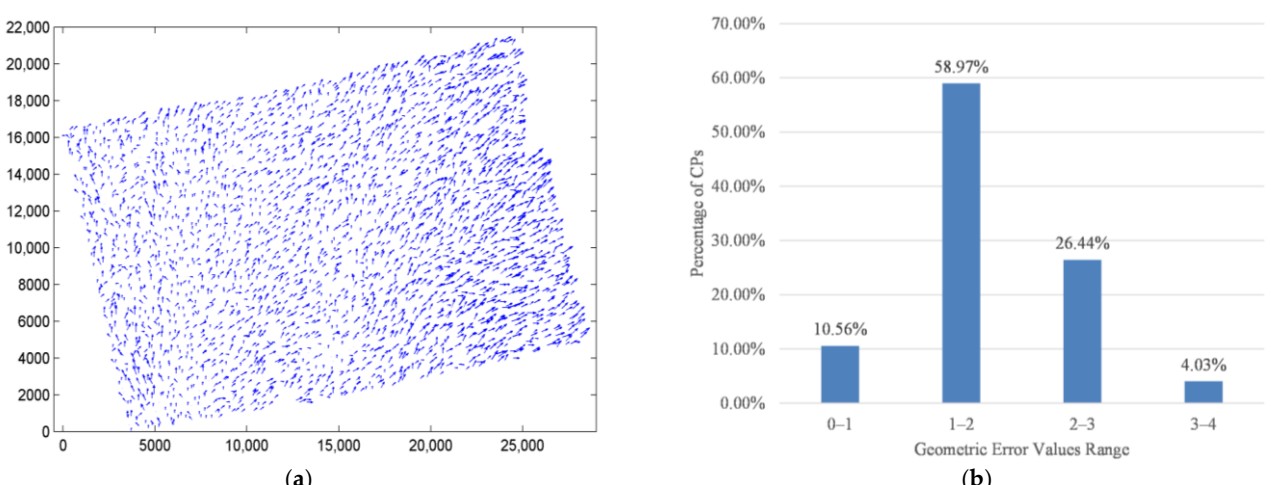

(**a**)                                    (**b**)

**Figure 25.** Analysis results of local geometric distortions in No. 2 image: (**a**) plot of geometric errors, (**b**) histogram of geometric error values.

The geometric error values of the CPs for the No. 3 image range from 0 to 4; there are significant differences in the geometric error directions of the CPs, but very small differences in the geometric error values of the CPs. Therefore, the No. 3 image has small local geometric distortions, as shown in Figure 26.

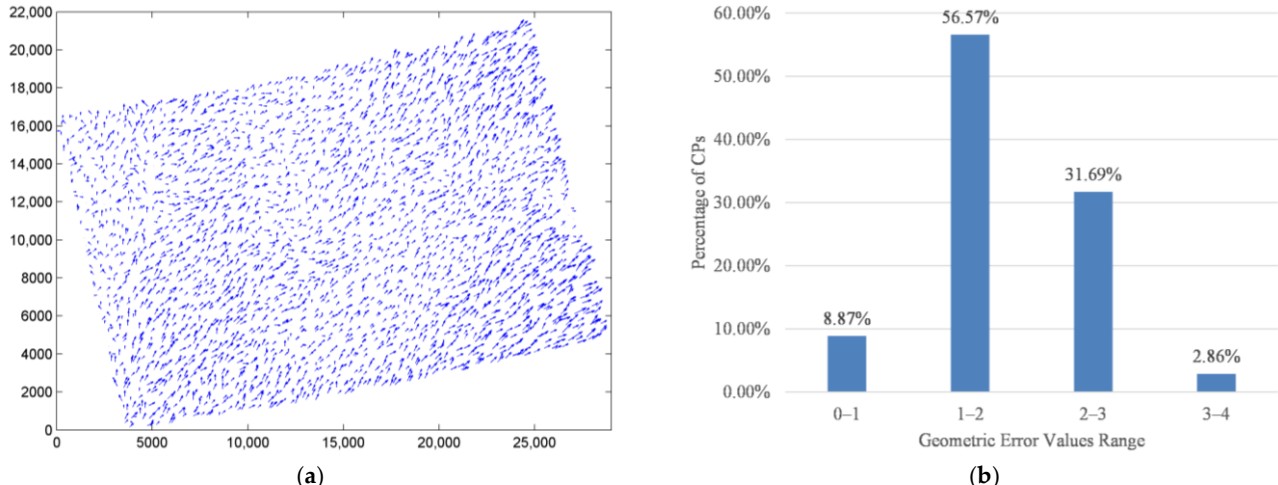

**Figure 26.** Analysis results of local geometric distortions in No. 3 image: (**a**) plot of geometric errors, (**b**) histogram of geometric error values.

The geometric error values of the CPs for the No. 4 image range between 1 and 5, and 85.38% of the CPs fall within a range of 2 to 4. There are significant differences in the geometric error directions of the CPs, but very small differences in the geometric error values of the CPs. Therefore, the No. 4 image has small local geometric distortions, as shown in Figure 27.

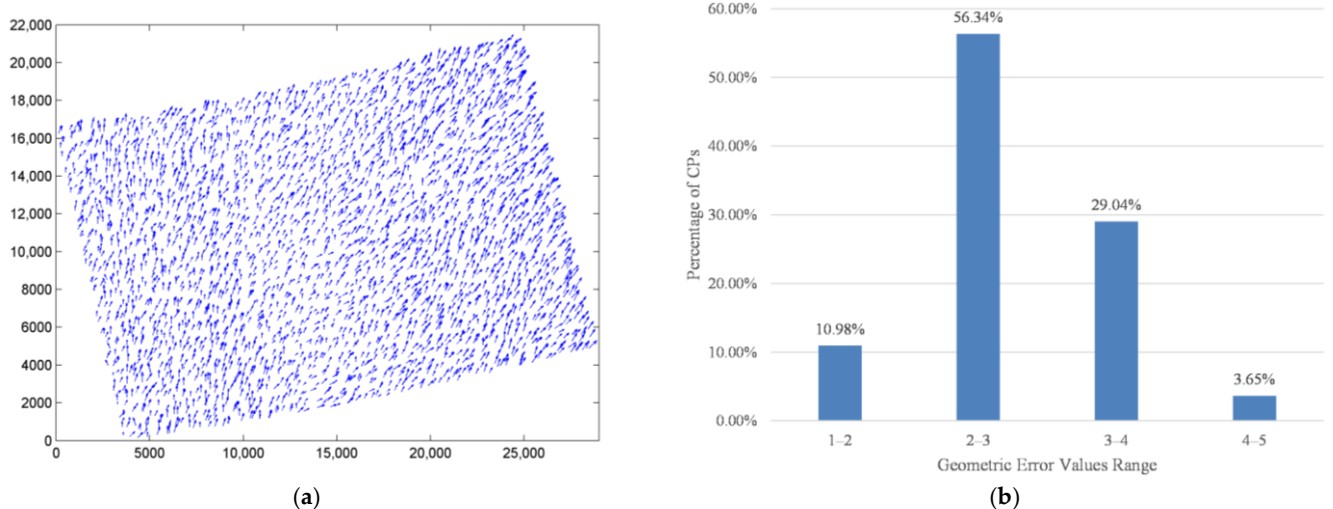

**Figure 27.** Analysis results of local geometric distortions in No. 4 image: (**a**) plot of geometric errors, (**b**) histogram of geometric error values.

The geometric error values of the CPs for the No. 5 image range from 15 to 19, with 85.96% of the CPs falling within a range of 16 to 18. There are small differences in the geometric error directions and values of the CPs. Therefore, the No. 5 image has very small local geometric distortions, as shown in Figure 28.

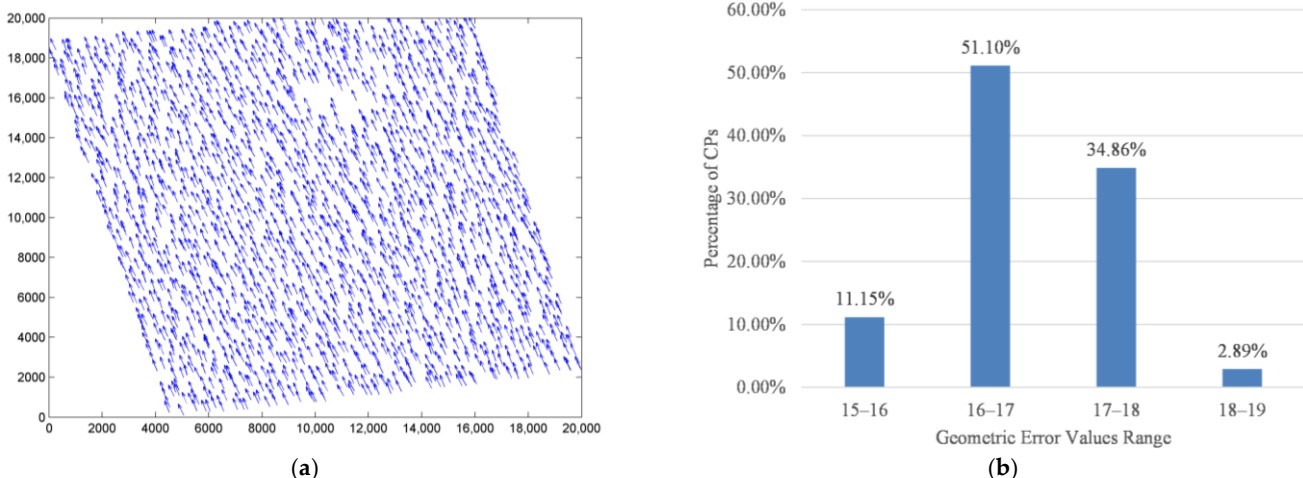

(**a**)　　　　　　　　　　　　　　　　　　　　　　　(**b**)

**Figure 28.** Analysis results of local geometric distortions in No. 5 image: (**a**) plot of geometric errors, (**b**) histogram of geometric error values.

The geometric error values of the CPs for the No. 6 image range from 1 to 4, with 91.4% of the CPs falling within a range of 2 to 3. There are significant differences in the geometric error directions of the CPs, but very small differences in the geometric error values of the CPs. Therefore, the No. 6 image has small local geometric distortions, as shown in Figure 29.

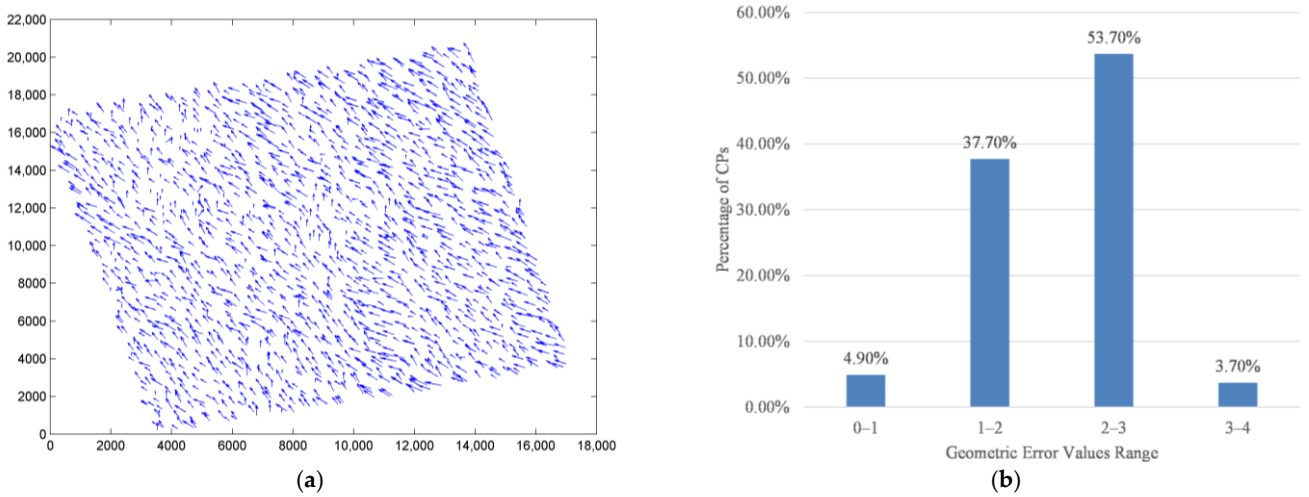

(**a**)　　　　　　　　　　　　　　　　　　　　　　　(**b**)

**Figure 29.** Analysis results of local geometric distortions in No. 6 image: (**a**) plot of geometric errors, (**b**) histogram of geometric error values.

The geometric error values of the CPs for the No. 7 image range from 4 to 11; there are small differences in the geometric error directions of the CPs, and relatively large differences in the geometric error values of the CPs. Therefore, the No. 7 image has relatively large local geometric distortions, as shown in Figure 30.

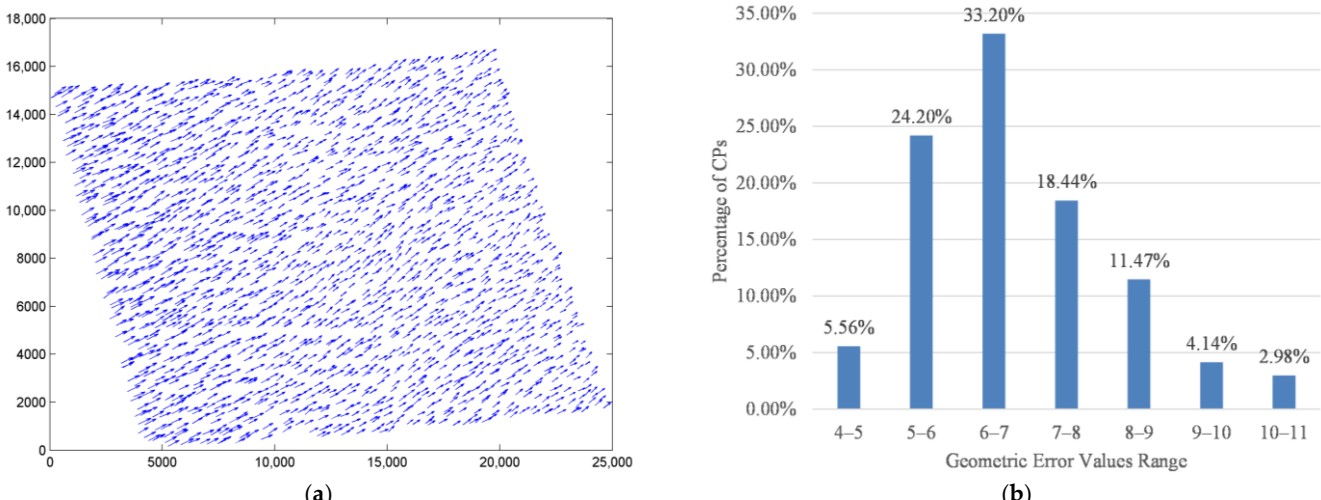

**Figure 30.** Analysis results of local geometric distortions in No. 7 image: (**a**) plot of geometric errors, (**b**) histogram of geometric error values.

The geometric error values of the CPs for the No. 8 image range from 0 to 3; there are significant differences in the geometric error directions of the CPs, but very small differences in the geometric error values of the CPs. Therefore, the No. 8 image has small local geometric distortions, as shown in Figure 31.

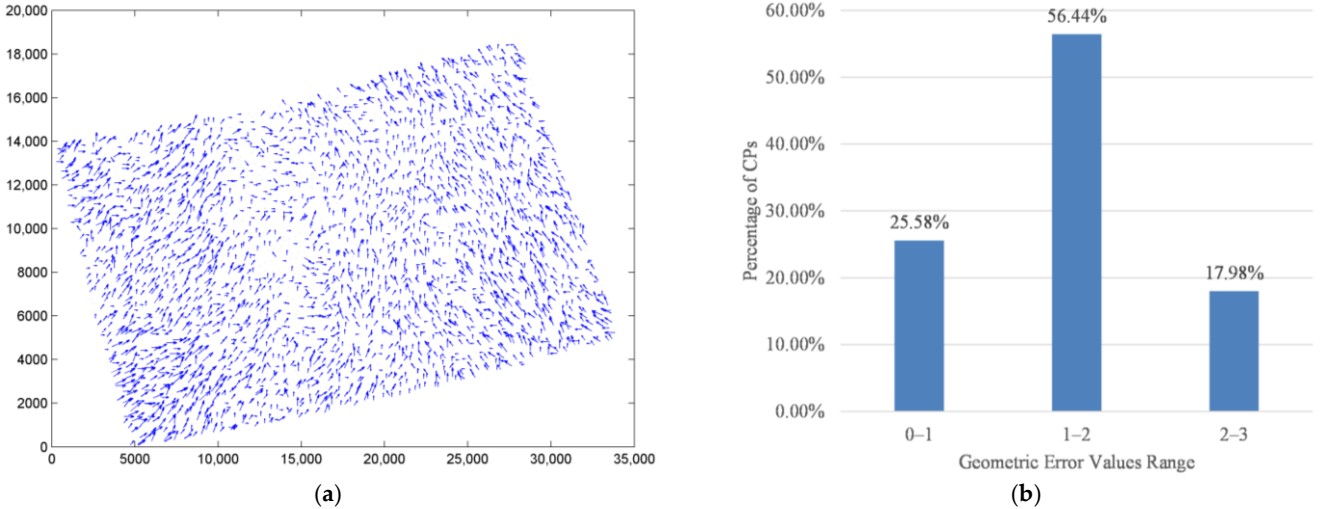

**Figure 31.** Analysis results of local geometric distortions in No. 8 image: (**a**) plot of geometric errors, (**b**) histogram of geometric error values.

The analysis of the local geometric distortions for the eight GF-6 images showed that the majority of the GF-6 images have significant differences in their geometric error directions, but very small differences in their geometric error values. Therefore, the majority of the images have small local geometric distortions.

Comparing the local geometric distortions of the GF-6 images and the GF-1 images showed that the local geometric distortions of the GF-6 images are smaller than those of the GF-1 images.

### 4.2. The Results for the Refined RFM

#### 4.2.1. Experimental Data

The experimental images consist of four GF-1 images and four GF-6 images; the details of the eight experimental images are shown in Table 5. Although most of the experimental

GF-1 images are in China, the experimental images have different geometric RFM accuracies and different features in different regions of China, which does not affect the geometric accuracy evaluation results.

**Table 5.** Experimental GF-1/GF-6 image information for geometric accuracy evaluation.

| Number | Sensors | Imaging Time | Country |
|--------|---------|--------------|---------|
| No. 1 | GF-1 WFV1 | 17 March 2014 | China |
| No. 2 | GF-1 WFV3 | 23 December 2018 | China |
| No. 3 | GF-1 WFV1 | 30 August 2019 | China |
| No. 4 | GF-1 WFV2 | 23 April 2021 | France |
| No. 5 | GF6 WFV | 26 January 2019 | China |
| No. 6 | GF6 WFV | 19 November 2020 | China |
| No. 7 | GF6 WFV | 14 September 2020 | Brazil |
| No. 8 | GF6 WFV | 5 March 2021 | Brazil |

### 4.2.2. Experimental Results

In this experiment, for each experimental image, multiple refined rational function models were constructed using different number of CPs; then, the geometric accuracy of each refined RFM was evaluated using the proposed method. The results of the geometric accuracy evaluation are shown in Table 6.

**Table 6.** Experimental results of refined RFM with different numbers of CPs.

| Number of CPs | RMSE | | | | | | | |
|---------------|-------|-------|-------|-------|-------|-------|-------|-------|
| | No. 1 | No. 2 | No. 3 | No. 4 | No. 5 | No. 6 | No. 7 | No. 8 |
| 0 | 78 | 2.3 | 7.1 | 4.7 | 1.9 | 1.9 | 16.8 | 2.2 |
| 6 | 3.12 | 1.42 | 1.59 | 1.43 | 1.19 | 1.09 | 1.21 | 0.98 |
| 12 | 2.98 | 1.14 | 1.46 | 1.42 | 0.97 | 1.06 | 1.16 | 0.97 |
| 16 | 2.71 | 1.16 | 1.45 | 1.38 | 0.96 | 0.98 | 1.13 | 0.95 |
| 20 | 2.73 | 1.12 | 1.42 | 1.33 | 0.93 | 0.95 | 1.14 | 0.91 |

For each image, the RMSE value decreases as the number of CPs increases, but the decrease in the RMSE value is not noticeable when the number of CPs increases to 20; the minimum value of the RMSE of each GF-1 image is greater than 1.0 and cannot meet the practical requirement of being better than 1.0 pixel. The reason for these results is that most GF-1 images have significant local geometric distortions, which cannot be corrected by a refined RFM. The minimum RMSE values of most GF-6 images is less than 1.0; however, the RMSE values are all greater than 0.9, and further processing is recommended to improve the geometric accuracy of the GF-6 images.

## 5. Conclusions

An automatic geometric accuracy evaluation method is proposed in this paper, which can be used to evaluate the geometric accuracy of the RFM for GF-1 and GF-6 images. First, RPC correction is completed using the RFM and refined RFM, respectively. The RFM is refined using some CPs, which results in a refined RFM. Second, an automatic matching method based on Harris, BRIEF, and template matching is proposed in this paper to obtain many evenly distributed CPs. SIFT and its improved methods can only obtain a few or no CPs in areas with inconspicuous features, and SIFT and most of its improved methods have a slow computational speed. The proposed method is superior to SIFT and BRIEF in terms of the number, distribution, and accuracy of the CPs, and the processing speed of the

proposed method is much better than that of SIFT. Finally, the RFM and refined RFM are evaluated using many GF-1 and GF-6 images.

The geometric accuracy of the RFM was evaluated using 10,561 GF-1/GF-6 images of China, Brazil, and France. The experimental results indicate that the geometric accuracy of the GF-1 images is mainly distributed between 0 and 20. In contrast, the geometric accuracy of the GF-6 images is mainly distributed between 0 and 7. The geometric accuracy of the GF-6 images is obviously superior to that of the GF-1 images. Local geometric distortion analysis was performed using eight GF-1 and GF-6 images. The experimental results showed that most GF-1 images have significant local geometric distortions, and most GF-6 images have small local geometric distortions. According to the results of the analysis, the RMSE values, and the local geometric distortions in GF-1/GF-6 images, the geometric accuracy of the GF-6 images is better than that of the GF-1 images. Therefore, the RFM accuracy of most GF-1 and GF-6 images does not meet the practical requirement of being better than 1.0 pixel. The reason for these results is that the satellite imaging parameters are inaccurate and the RFM is not optimized using CPs.

The accuracy of the refined RFM was evaluated using four GF-1 images and four GF-6 images. The accuracy of the refined RFM of all the GF-1 experimental images does not meet the practical requirements, and the reason for these results is that most GF-1 images have significant local geometric distortions, which cannot be corrected by a refined RFM. When 20 CPs are used to construct a refined RFM, the accuracy of the refined RFM of most GF-6 experimental images can meet the practical requirement of being better than 1.0 pixel; however, the RMSE values are all greater than 0.9, and further processing is recommended to improve the geometric accuracy of the GF-6 images. In future work, the automatic matching method will be used to obtain CPs to refine the RFM of GF-1 and GF-6 images, so that an accuracy evaluation of a large number of images can be completed. The image registration method should be used to further improve the geometric accuracy of GF-1/GF-6 images.

The geometric accuracy evaluation results presented in this paper can help researchers to develop a high-precision geometric processing algorithm for GF-1/GF-6 images. The proposed method can also be used to evaluate the geometric accuracy of other Chinese satellite images.

**Author Contributions:** Conceptualization, X.S.; methodology, X.S.; software, X.S.; validation, J.Z.; formal analysis, J.Z.; resources, J.Z.; data curation, J.Z.; writing—original draft preparation, X.S.; writing—review and editing, X.S.; visualization, X.S. and J.Z. All authors have read and agreed to the published version of the manuscript.

**Funding:** This research was funded by the National Key Research and Development Program of China (grant number 2019YFE0197800).

**Data Availability Statement:** Not applicable.

**Conflicts of Interest:** The authors declare no conflict of interest.

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
