# Peer review of "Does the Rational Function Model’s Accuracy for GF1 and GF6 WFV Images Satisfy Practical Requirements?"

_remotesensing, doi:10.3390/rs15112820_

Round 1
Reviewer 1 Report (Previous Reviewer 2)
The authors have taken into consideration the comments of the reviewer and improved the original manuscript sufficiently. A few suggestions for further improvement are as follows.
Lines 16, 61, 122, etc.: The expression "Refined RFM is refined" is misleading, because the refined RFM is not refined again, but the RFM is refined which results in the refined RFM. So, it should be "RFM is refined".
Line 264: "in descending order" should be "in ascending order".
Figures 10-15: The vertical axis label should be "Percentage of images" instead of "RMS Percentage".
Figures 16b-31b: The vertical axis label should be "Percentage of CPs" instead of "Geometric Error Values Percentage".
Section 4.1.3: It would be better to analyze the local geometric distortions using the refined RFM than the RFM. The distortions would be then better visible when systematic shifts between the images had been corrected.
Missing references to be cited and discussed:
A geometric accuracy of sub-pixel level (better than in this manuscript) has been reported for GF-6 images, using a RFM with systematic errors compensated by an affine transformation between corresponding GCPs (similar method as in this manuscript), in
Yin, S.; Zhu, Y.; Hong, H.; Yang, T.; Chen, Y.; Tian, Y. DSM Extraction Based on Gaofen-6 Satellite High-Resolution Cross-Track Images with Wide Field of View. Sensors 2023, 23, 3497. https://doi.org/10.3390/s23073497
Zhao, L., Fu, X., Dou, X. (2020). Analysis of Geometric Performances of the Gaofen-6 WFV Camera. In: Wang, L., Wu, Y., Gong, J. (eds) Proceedings of the 6th China High Resolution Earth Observation Conference (CHREOC 2019). CHREOC 2019. Lecture Notes in Electrical Engineering, vol 657. Springer, Singapore. https://doi.org/10.1007/978-981-15-3947-3_8
Zhao, L., Fu, X. (2020). Evaluation and Analysis of Geometric Performances of the Gaofen-1 B/C/D Satellite. In: Wang, L., Wu, Y., Gong, J. (eds) Proceedings of the 6th China High Resolution Earth Observation Conference (CHREOC 2019). CHREOC 2019. Lecture Notes in Electrical Engineering, vol 657. Springer, Singapore. https://doi.org/10.1007/978-981-15-3947-3_9
A paper for compensation of image distortions:
Mi Wang, Yufeng Cheng, Beibei Guo, Shuying Jin, Parameters determination and sensor correction method based on virtual CMOS with distortion for the GaoFen6 WFV camera, ISPRS Journal of Photogrammetry and Remote Sensing, Volume 156, 2019, Pages 51-62. https://doi.org/10.1016/j.isprsjprs.2019.08.001
There are mistakes in English but the text is understandable.
Author Response
Dear Reviewer:
Thanks very much for taking your time to review this manuscript. I really appreciate all your comments and suggestions. We have made corrected modification on the revised manuscript. This manuscript had been reviewed and edited by language of services of MDPI.

Reviewer 2 Report (New Reviewer)
The reviewer would like to thank the authors for this thoughtful manuscript. This work has good potential. The authors are requested to put in some additional efforts to improve the quality of this manuscript.
Introduction
The authors are requested to elaborate on how Gaofen can help in early response of natural disaster related events. Please cite the following article that reports a major disaster and discuss how post-disaster analysis of such an event can benefit from the data in terms of loss of inhabited area assessment through post & pre-disaster change comparison. The event largely impacted the local human livelihoods as well as manmade infrastructure.
i) Shugar et al, A massive rock and ice avalanche caused the 2021 disaster at Chamoli, Indian Himalaya, Science, 2021.
Accuracy Analysis
The authors are requested to discuss the effectiveness of performance metrics (like RMSE etc.) as demonstrated in the following interdisciplinary articles and discuss the use of these metrics for geometric accuracy analysis.
i) Hastie et al., 2009. The elements of statistical learning: Data Mining, Inference, and Prediction
ii) Muhuri et al., 2021. Performance Assessment of Optical Satellite-Based Operational Snow Cover Monitoring Algorithms in Forested Landscapes, IEEE JSTARS.
Barplots
The authors are requested to merge the barplots in Fig. 10,11,12 and so on to be able to better compare the performance.
DEM Based Comparison
The authors have considered most of the test sites in China and only one in France. Please explain how the proposed technique will behave as a function of topography.
Conclusion
The authors are requested to list the key contributions in this section. At the moment the section is not detailed enough.
A grammatical check and spell check is requested.
Author Response
Dear Reviewer:
Thanks very much for taking your time to review this manuscript. I really appreciate all your comments and suggestions. We have made corrected modification on the revised manuscript. This manuscript had been reviewed and edited by language of services of MDPI

This manuscript is a resubmission of an earlier submission. The following is a list of the peer review reports and author responses from that submission.
Round 1
Reviewer 1 Report
This paper evaluates the RPC accuracy of GF-1 and GF-6 WFV sensors using a large number of L1-level images. The conclusion drawn is that the RPC accuracy of these sensors does not meet practical requirements. However, the paper is poorly prepared and contains numerous grammatical and typographical errors. Additionally, the paper does not introduce any new techniques to the research community, and therefore, it should be classified as a communication report rather than a research study.
More critically, the approach utilized in this paper to evaluate geometric precision is flawed, which casts doubt on the validity of the conclusions. Therefore, I suggest that this paper be rejected.
Detailed comments are as follows:
1. Matching the RPC-corrected images with Google Earth images alone may not be sufficient to evaluate the accuracy of the vendor-provided RFM. It is recommended to perform terrain correction as well to obtain a more comprehensive assessment.
2. It is important to note that local geometric distortions do not necessarily reflect the absolute geolocating accuracy. Therefore, even if the "RMSE" is high, the local geometric distortion may still be minimal.
3. The L1-level product is not ready for orthorectification and usually requires ground control points to correct the bias in the geometric model. This is a common practice among sensors and should be taken into account when processing the L1-level product.
4. “GF-16” appeared many times in the manuscript.
5. Consistent and accurate use of terminology is important in scientific writing. It may be helpful to review the manuscript and ensure that the terms "China", "Chinese", "France", "French", "Brazil", and "Brazilian" are used consistently and appropriately throughout.
6. If there is a discrepancy between the description of the workflow and what is shown in Figure 2, it may be necessary to revise one or the other to ensure that they are consistent. If "Template matching" should only be used when BRIEF matching fails, this should be clarified in the manuscript text or in a revised figure caption.
Reviewer 2 Report
Brief summary:
The manuscript evaluates the geometric accuracy of GF-1 and GF-6 satellite images against Google Earth images. The GF-1/GF-6 image coordinates are transformed using a set of rational functions as a camera model and a DEM. Corresponding points between the GF-1/GF-6 images and Google Earth images are established using feature points detected by the Harris operator and described by the BRIEF descriptor. Incorrectly matched feature points are re-matched using fast normalized cross-correlation. There are differences of several pixels between the corresponding points in the experimental results. It is claimed that the differences are due to a low accuracy of the rational function model.
General comments:
1) It has not been verified that the reason for the low accuracy is in the rational function model. The reason can also be in the feature points, DEM, or Google Earth images which have been captured from different viewpoints than the GF-1/GF-6 images. The rational function model is often refined with ground control points (see Hu et al. (2004) below), which has not been done in the manuscript and can be the main reason for systematic errors visible in the results.
2) The authors use fast computation as a criterion in their approach but do not present any computing times.
3) It is misleading to state that the approach is based on "multi-feature" matching. The feature points are detected by the Harris operator and described by the BRIEF descriptor. This is only a single feature. Template matching is not a feature-based method but an area-based one. It is not a feature.
4) There are many similar figures showing the histograms and directions of errors. It would be good to present these more compactly and show also an example of an original GF-1/GF-6 image, a Google Earth image, and matched points between these images.
5) A discussion of the results is missing, including possible reasons for the low accuracy and suggestions for the future research to improve the accuracy.
6) All the references except one are older than five years, but adequate anyway. More recent and missing references include the following.
Nwilo, P.C., Okolie, C.J., Onyegbula, J.C. et al. Positional accuracy assessment of historical Google Earth imagery in Lagos State, Nigeria. Appl Geomat 14, 545–568, 2022. https://doi.org/10.1007/s12518-022-00449-9
Calonder, M., Lepetit, V., Strecha, C., Fua, P. (2010). BRIEF: Binary Robust Independent Elementary Features. In: Daniilidis, K., Maragos, P., Paragios, N. (eds) Computer Vision – ECCV 2010. ECCV 2010. Lecture Notes in Computer Science, vol 6314. Springer, Berlin, Heidelberg. https://doi.org/10.1007/978-3-642-15561-1_56
Hu, Y., Tao, V., and Croitoru, A. Understanding the rational function model: Methods and applications. International Archives of Photogrammetry and Remote Sensing 20(6), 119-124, 2004.
Detailed comments:
Lines 8-20, 51-53, 90-94: There is mixed use of GF-16 and GF-6 in the abstract and Section 1. It seems that GF-16 should be GF-6.
Lines 11-12 and throughout the manuscript: Is orthorectification also carried out? Have Google Earth images been orthorectified? In Google Earth, the images are from a tilted viewpoint and most likely from a different viewpoint than the GF-1/GF-6 images.
Lines 12, 60, 371, and 376: Precision describes the spread of measurements around their mean, while accuracy includes also possible systematic errors. It is not clear why there is precision on these lines instead of accuracy.
Lines 30 and 36-37: The units of the spectral ranges are missing.
Line 36: Is the spectral range of 0.45 - 0.45 correct (a narrow range compared to the other ranges)?
Line 47: Please write out the acronym OGC.
Line 75: Either reference to [8] is incorrect or BRIEF should be BRISK. [8] deals with BRISK while BRIEF is proposed in Calonder et al. (2010) given above.
Lines 79-84: It reads that SIFT and SURF do not produce control points "in areas with inconspicuous features", but template matching does, because it includes "more informative regional grayscale information". Is this claim based on that you use a large window for the template matching while SIFT and SURF use only local information in a e.g. 16 x 16 window around the feature point?
Line 88: Please include a reference to BRISK [8] here or on line 75.
Line 93: It is not clear what "14-level" means.
Line 114: Please clarify which DEM of 90 m resolution is used in the image coordinate transformation. What is the accuracy of the DEM? How does the inaccuracy of the DEM propagate to the result of the coordinate transformation?
Lines 119-122 versus Abstract and Introduction: It reads that the GF-1/GF-16 images are matched with the Google Earth images for the first time on lines 119-122, while this is not clear in the abstract and introduction. Please mention already in the abstract and introduction that the matching is carried out against reference images which are Google Earth images in this case.
Line 148: Here is confusion between BRISK and BRIEF again. Do you use BRISK or BRIEF?
Lines 149-151: BRIEF is based on intensity differences at test locations in a region, so it builds the descriptor from data in a region like the conventional methods.
Line 168: The value of B has not been given in the experiments.
Lines 172-173: It is not clear what "expanded proportionally" means.
Line 184: Reference to [12] is incorrect. Fast NCC is presented in [13].
Line 186: It is not clear whether 400x400 is the size of the grid or the size of a cell in a grid.
Line 187: Please clarify how the optimal matching degree is defined. Optimal in which sense?
Lines 167-188: It would be clearer if the feature point detection (Harris) and outlier detection (RANSAC) steps were also included in the description of the algorithm.
Line 218: Please specify what would be a required accuracy in practical applications.
Captions of Figs. 9a-24a: The figures show the directions and not distributions of the geometric errors.
Line 277 and Fig. 13b: There is a contradiction: It reads in the text that GEV is between 8-13 but in the figure, it is between 9-13.
Line 282 and Fig. 14b: There is a contradiction: It reads in the text that GEV is between 9-13 but in the figure, it is between 8-13.
Lines 311-320 and Figs. 17-18: The spread of GEVs and error directions is similar in Fig. 18 as in Fig. 17, so the distortions are similar (either significant or small) in both cases.
Line 326: Image No. 2 should be No. 3.
Line 332: "... as shown in Figure 19" should be "... as shown in Figure 20".
Line 338: "... as shown in Figure 19" should be "... as shown in Figure 21".
Line 356: "... as shown in Figure 22" should be "... as shown in Figure 24".
Lines 369-370: The accuracy of the CPs has not been verified, so we do not know if they meet the accuracy requirement in addition to being uniformly distributed.
Line 411: The link to [13] has an extra empty space and don't work because of it.
Reviewer 3 Report
This paper is more experimental report rather comperhisenive research paper. I cannot find any novality, and several components of proper research need to be included.
*The author should provide only relevant information related to this paper and reserve more space for the proposed framework.
*However, the author should compare the proposed algorithm with other recent works or provide a discussion. Otherwise, it's hard for the reader to identify the novelty and contribution of this work.
*The descriptions given in this proposed scheme areneed to be morefficient that this manuscript only adopted a variety of existing methods to complete the experiment where there are no strong hypothesis and methodical theoretical arguments. Therefore, the reviewer considers that this paper needs more works.
*Key contribution and novelty has not been detailed in the manuscript. Please include it in the introduction section
*What are the limitations of the related works
*Are there any limitations of this carried out study?
*How to select and optimize the user-defined parameters in the proposed model?